# Differential syntactic and semantic encoding in LLMs

**Santiago Acevedo** [1]   **Alessandro Laio** [1]   **Marco Baroni** [2]

## Abstract

We study how syntactic and semantic information is encoded in inner layer representations of Large Language Models (LLMs), focusing on the very large DeepSeek-V3. We find that, by averaging hidden representations of sentences sharing syntactic structure or meaning, we obtain vectors that capture a significant proportion of the syntactic and semantic information contained in the representations. In particular, subtracting these syntactic and semantic "centroids" from sentence vectors strongly affects their similarity with syntactically and semantically matched sentences, respectively, suggesting that syntax and semantics are, at least partially, linearly encoded. We also find that the cross-layer encoding profiles of syntax and semantics are different, and that the two signals can to some extent be decoupled, suggesting differential encoding of these two types of linguistic information in LLM representations.

## 1. Introduction

The success of LLMs has spurred wide interest in decoding where and how information is stored in their high-dimensional representations, both to improve and better control AI systems (Ferrando et al., 2024; Zhao et al., 2024), and to address the fundamental scientific question of how high-dimensional, vector-based models store linguistic competence traditionally assumed to be symbolically represented (Futrell & Mahowald, 2025; Levy et al., 2025). Two core aspects of linguistic competence are *syntax*, pertaining to the structural scaffolding of sentences, and *semantics*, that is, the *meaning* denoted by the sentences. While all linguists recognize the central role of syntax and semantics in language, there has been much disagreement about how strictly separated the two components are, with the generative tradition being associated with a strong stance on the autonomy of syntax (e.g., Chomsky, 1965; 1986), whereas functionalist approaches regard the two components as strongly entangled (e.g., Croft & Cruse, 2004; Goldberg, 2019).

We contribute here two main results relevant to these debates. First, we show that syntax and semantics are at least partially represented in the inner layers of LLMs through a simple linear encoding scheme. The syntactic and semantic information present in a sentence can be approximated by averaging sentences sharing the same syntactic structure or semantic contents, obtaining what we call syntactic and semantic *centroids*. The same information can then be removed by subtracting (a linear function of) the respective centroids from sentence vector representations. Second, we find that syntax and semantics can be partially separated in LLM representations, as shown by two types of evidence: i) The layers in which the syntactic and semantic components more strongly characterize sentence vectors only partially overlap, with semantics most clearly encoded in the central layers of the network, whereas syntax remains salient for a wider range of layers. ii) Strikingly, removing the semantic centroid from a sentence vector does not significantly affect its syntactic contents. Removing the syntactic centroid produces instead stronger effects on measurements of semantic information, suggesting an asymmetry in how syntax and semantics influence each other.

From the LLM interpretability angle, our results contribute to the recent literature showing that some form of deeper linguistic processing is taking place in the central layers of LLMs (Cheng et al., 2025; Skean et al., 2025). We suggest that this processing more specifically pertains to semantics. It also brings further evidence for the hypothesis that simple linear superposition works as one of the fundamental mechanisms by which deep nets encode information (Mikolov et al., 2013; Park et al., 2024; 2025), showing that this encoding is also used for abstract features such as the syntactic structure and the meaning of a sentence. From a linguist perspective, we observe, in LLMs trained without explicit linguistic priors, the emergence of an imperfect but clear separation between syntax and semantics, suggesting there might be something inherently optimal in an encoding of language that makes this distinction.

[1]Scuola Internazionale Superiore di Studi Avanzati (SISSA), Trieste, Italy [2]Catalan Institute of Research and Advanced Studies (ICREA) and Universitat Pompeu Fabra (UPF), Barcelona, Spain. Correspondence to: Santiago Acevedo <sacevedo@sissa.it>, Alessandro Laio <laio@sissa.it>, Marco Baroni <marco.baroni@upf.edu>.

*Proceedings of the 43ʳᵈ International Conference on Machine Learning*, Seoul, South Korea. PMLR 306, 2026. Copyright 2026 by the author(s).

*Table 1.* **Examples of syntax matching.** Pairs of sentences sharing syntax (as cued by POS templates) but expressing unrelated meanings. POS tags are from the Penn Treebank tagset, see Appendix A.1.

| Original sentences $X_i$ | Syntax twins $s_i^0$ |
| --- | --- |
| The weary traveler stopped at an inn | A clever fox hid behind a bush |
| DT JJ NN VBD IN DT NN | |
| They were constantly monitored within the system | He was kindly remembered throughout the town |
| PRP VBD RB VBN IN DT NN | |
| He wrote a long letter yesterday | I met a famous person once |
| PRP VBD DT JJ NN RB | |
| We will show off the new invention | They must draw up a detailed plan |
| PRP MD VB RP DT JJ NN | |

## 1.1. Related work

Recent work has found representations in the deeper layers of LLMs to align across models and related stimuli, suggesting that deep layers are performing some kind of shared processing, presumably abstract and linguistic in nature (e.g., Antonello & Cheng, 2024; Huh et al., 2024; Peng & Søgaard, 2024; Acevedo et al., 2025; Brinkmann et al., 2025; Cheng et al., 2025; Lee et al., 2025; Lindsey et al., 2025; Wolfram & Schein, 2025). However, *what kind* of linguistic processing this is remains an open question. Concurrent literature has used various "probes" to check where LLM inner representations process information related to syntax, semantics and other linguistic levels (e.g., Hewitt & Manning, 2019; Jawahar et al., 2019; Tenney et al., 2019; Niu et al., 2022; He et al., 2024; Simon et al., 2024; Li & Subramani, 2025; Simon et al., 2025). The emergent consensus is that LLMs reproduce a classic linguistic pipeline in which the deep layers perform syntactic analysis followed by semantic processing, a result we partially confirm here using our complementary geometric methods to probe syntactic and semantic information. More generally, Mahowald et al. (2024) differentiate formal linguistic competence from more functional aspects of language use both in LLMs and brains.

A different research line has investigated the extent to which LLMs encode information with simple linear encoding schemes (e.g., Mikolov et al., 2013; Hernandez et al., 2024; Park et al., 2024; Korchinski et al., 2025; Park et al., 2025). Like us, Caucheteux et al. (2021) use the average of GPT2 sentence representations with the same syntactic structure as a proxy for the syntactic information contained in them. They use this representation to try to disentangle brain responses to syntax and semantics in a LLM-to-brain encoding setup.

We build here on this existing literature by establishing a partial syntax/semantics dissociation in LLM representations using only linear methods.

## 2. Data and Methods

### 2.1. Syntactic and semantic datasets and centroids

#### 2.1.1. MATCHED-SENTENCE DATASETS

Our study is based on comparing pairs of sentences that are matched in terms of either syntactic or semantic information, and measuring how the similarity between their representations is affected by various ablations. Syntactically matched pairs were generated using Gemini and ChatGPT, two LLMs that were not further used for analysis. In particular, we built pairs of sentences that had the same syntactic profile, defined as the same sequence of parts of speech (POSs),[1] but different meanings. We collected roughly 2,000 such pairs. See Appendix A.1 for a detailed description of how they were generated. For each syntactically matched pair, one item is the *original sentence*, $X_i$, and we call the other item its *syntax twin*, $s_i^0$. The set of original sentences and the set of syntax twins have no elements in common, and there are multiple distinct pairs of sentences sharing the same POS template. Table 1 provides some examples.

On the semantics side, for each original sentence $X_i$, we generated an English paraphrase $P_i$ using ChatGPT. See Table 2 for an example, and Appendix A.2 for details.

---

[1]POS templates could in principle correspond to different syntactic structures: "I ate the mango with a spoon" and "I liked the twist in the plot" have identical POS sequences (PRP VBD DT NN IN DT NN) but different structures (the prepositional phrase "with a spoon" modifies the verb phrase "ate the mango" in the first sentence, whereas the prepositional phrase "in the plot" modifies the noun phrase "the twist" in the second). We manually checked sample examples of all our POS templates, finding that, while some are potentially ambiguous, in practice they tend to instantiate a single syntactic structure. Using POS templates, which can be more robustly identified than syntactic parses, implies that we will not be comparing alternate syntactic structures that map to the same POS sequence. However, importantly, different POS templates do pick up different syntactic structures, which is the only condition we need to satisfy for our experimental design to work.

*Table 2.* **Definitions with examples.** $N_{\text{twins}}$ stands for the number of sentences sharing the same POS structure, and $N_{\text{languages}} = 6$ is the number of languages in our dataset (Chinese, Spanish, Italian, Turkish, German and Arabic). For details about the dataset, see Sec. 2.1 and appendices A.1, A.2 and A.3. Note that the syntax centroid $\mathbf{S}_i$ does not include $\mathbf{X}_i$. Similarly, the semantic centroid $\mathbf{T}_i$ does not contain $\mathbf{X}_i$ nor $\mathbf{P}_i$.

| | | |
|---|---|---|
| **Original sentence** | *We must carefully observe the rules* | $\mathbf{X}_i$ |
| **Paraphrase** | *The rules must be followed with care* | $\mathbf{P}_i$ |
| **Syntax twins** 
 Sentences with the 
 same POS template 
 of $\mathbf{X}_i$: 
 PRP MD RB VB DT NNS | *We can barely see the stars* 
 *You could neatly arrange the items* 
 *We should fairly distribute the tasks* 
 *She will kindly assist the newcomers* 
 . . . | $\mathbf{s}_i^0$ 
 $\mathbf{s}_i^1$ 
 $\mathbf{s}_i^2$ 
 $\mathbf{s}_i^3$ 
 . . . |
| **Syntax centroid** | | $\mathbf{S}_i = \sum_{\alpha=0}^{N_{\text{twins}}} \mathbf{s}_i^\alpha / N_{\text{twins}}$ |
| **Translations** | *Debemos observar cuidadosamente las reglas* 
 *Dobbiamo osservare attentamente le regole* 
 *Kurallara dikkatle uymalıyız* 
 *Wir müssen die Regeln sorgfältig beachten* 
 . . . | $\mathbf{t}_i^{es}$ 
 $\mathbf{t}_i^{it}$ 
 $\mathbf{t}_i^{tr}$ 
 $\mathbf{t}_i^{de}$ 
 . . . |
| **Semantic centroid** | | $\mathbf{T}_i = \sum_{\gamma \in \{languages\}} \mathbf{t}_i^\gamma / N_{languages}$ |
| **Syntax ablation** | | $\mathbf{X}_i \perp \mathbf{S}_i = \mathbf{X}_i - \dfrac{\mathbf{X}_i \cdot \mathbf{S}_i}{|\mathbf{S}_i|^2} \mathbf{S}_i$ |
| **Semantic ablation** | | $\mathbf{X}_i \perp \mathbf{T}_i = \mathbf{X}_i - \dfrac{\mathbf{X}_i \cdot \mathbf{T}_i}{|\mathbf{T}_i|^2} \mathbf{T}_i$ |

### 2.1.2. SYNTACTIC AND SEMANTIC CENTROIDS AND THEIR ABLATION

In order to ablate the syntactic or semantic components from the paired sentences, we need to obtain a representation of them that abstracts away from other information. The representation of a specific sentence generated by an LLM on a certain layer is a high-dimensional real-valued vector where neurons have no predefined or architectural role, and thus there is no prior knowledge on how semantics or syntax are encoded in neural activity. However, one could expect representations of a collection of sentences with the same syntactic structure to have a partially *shared* pattern of neural activity. If this is the case, the *mean* neural activity among such sentences should retain to some extent the shared information and average out the rest.

Following this intuition, we first construct *syntactic centroids* by averaging representations of sentences sharing a POS template. More precisely, given that we have several instances of pairs of sentences sharing each POS template, for each original sentence $\mathbf{X}_i$ we gather all the syntax twins that share its POS template, that we call $\mathbf{s}_i^j$ with $j \geq 0$, and we average them to construct the syntactic centroid vector, $\mathbf{S}_i$. Since the meaning varies across the twins, semantic effects are, at least approximately, "averaged-out", whereas syntax information remains.

In Sec. 3.1, we will ablate syntactic information from $\mathbf{X}_i$ by subtracting its projection along the direction of its syntactic centroid. Formally, this operation, that makes $\mathbf{X}_i$ orthogonal to $\mathbf{S}_i$, can be expressed as

$$\mathbf{X}_i \rightarrow \mathbf{X}_i \perp \mathbf{S}_i = \mathbf{X}_i - \frac{\mathbf{X}_i \cdot \mathbf{S}_i}{|\mathbf{S}_i|^2} \mathbf{S}_i. \tag{1}$$

In order to extract the semantic component of a sentence in an analogous way to what we did for syntax, we first obtain its translations into 6 languages (Arabic, Chinese, German, Italian, Spanish and Turkish), using Gemini, Chat-GPT and, for Chinese only, DeepSeek (see examples in Table 2 and further details in Appendix A.3). We define $\mathbf{T}_i$, the *semantic centroid* of sentence $\mathbf{X}_i$, as the average of the representations of all translations (thus excluding the English original sentence $\mathbf{X}_i$ and its English paraphrase $\mathbf{P}_i$). We ablate the semantic information from $\mathbf{X}_i$ and $\mathbf{P}_i$ by subtracting their respective projections along $\mathbf{T}_i$, analogously to Eq. (1) above.

Syntactic centroids $\mathbf{S}_i$ and semantic centroids $\mathbf{T}_i$ are built with sentences which do not belong to the set of original sentences $\{\mathbf{X}_i, \ i = 1, \dots \}$. Therefore, the ablation operations on $\mathbf{X}_i$ do not involve any potential neighbor sentence $\mathbf{X}_j$. This procedure is designed to avoid spuriously strong ablation effects, in which one removes components of the neighbors.

The structure of our datasets and the operations involved in building centroids and using them for ablation are summarized in Table 2.

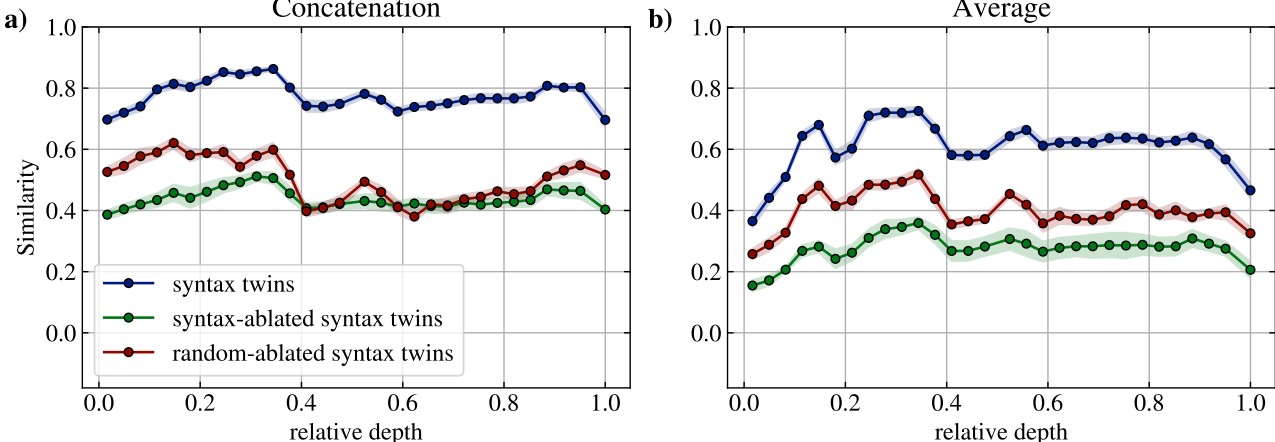

*Figure 1.* **Syntax similarity and its ablation**. Similarity between equal-syntax sentences (syntax twins), such as those presented in Table 1. Panels a) and b) represent sentences by token concatenation and average, respectively. The shaded colored areas represent 1 standard deviation, calculated by subsampling five times half of the samples.

## 2.2. Representation aggregation

As we are analyzing whole sentences, the issue arises of how to aggregate the hidden representations of their multiple tokens into a single vector. Following Acevedo et al. (2025), we consider two methods: either *concatenating* the representations of the last $N$ tokens, or *averaging* them. Given that measuring the similarity between concatenations requires using the same number of vectors across the board, we fix $N$ to be the minimum number of tokens in each dataset (6 for syntax and 3 for semantics).[2]

Note that another popular aggregation method is to use the last token of the sentence to represent it as a whole (e.g., Yin et al., 2024; Cheng et al., 2025). Acevedo et al. (2025) show that the resulting representation is essentially an impoverished version of the ones obtained by either averaging or concatenation. We informally experimented with last-token representations, finding similar patterns to those obtained with the other aggregation methods, but a less pronounced signal.

## 2.3. Representation similarity

Given the vector representations of two data points, several statistics can be used to quantify how similar they are (Klabunde et al., 2025; Sucholutsky et al., 2025). Following recent literature, we measure syntactic or semantic similarity between two representations generated by LLMs by comparing their high-dimensional neighborhoods. This approach was found to be preferable to linear similarity metrics such as Centered Kernel Alignment (CKA) (Korn-

---

[2]We replicated the semantic experiments with $N = 6$ on the data subset for which we had at least 6 tokens per sentence, obtaining essentially the same results.

blith et al., 2019), which provided very weak signals in high dimensional settings (Bansal et al., 2021; Huh et al., 2024; Acevedo et al., 2025; Cheng et al., 2025).

Given two representation spaces $A$ and $B$, for example those generated by a network processing the original sentences $\mathbf{X}_i$ and their syntax twins $\mathbf{s}_i^0$, respectively, we quantify their *similarity* by computing the average rank in a representation of the nearest neighbors of the other representation:

$$\text{Similarity} = 1 - \frac{1}{N_s^2} \left( \sum_{i,j:r_{ij}^A=1} r_{ij}^B + \sum_{i,j:r_{ij}^B=1} r_{ij}^A \right) \quad (2)$$

where $N_s$ is the number of samples, $r_{ij}^A$ is the distance rank of data point $i$ with respect to data point $j$ in representation A. Note that this measure takes a value of $0$ if the two representations are independent, and $1$ if the first neighbors in both spaces coincide. Moreover, it is invariant under global translations, global rotations, and global scalings of data, since it is based on distance ranks between data points.

Our similarity measure is closely related to the Information Imbalance, introduced in Glielmo et al. (2022), and used to analyze neural network representations by Cheng et al. (2025); Acevedo et al. (2025), and to the Neighborhood Overlap from Huh et al., 2024. For further details, see Appendix B.

## 2.4. Models

The results reported in the main text are obtained with DeepSeek-V3 (DeepSeek-AI et al., 2025), with 671b parameters. Appendix J shows that all our results are qualitatively reproduced with the much smaller Qwen2-7b (Qwen Team, 2024) and Gemma3-12b (Gemma Team, 2025), two models

that support all languages we work with. In Appendix H, we use Pythia6.9b (Biderman et al., 2023), as it allows us to look at the progress of representations during training.

## 3. Results

### 3.1. Syntactic similarity

Fig. 1 shows the similarity between our original sentence set and their respective syntax twins (equal-syntax sentences), using either concatenated or averaged sentence representations.[3]

For concatenated tokens (panel a)), we observe high similarity values, greater than 0.7 throughout *all* layers of the network. Panel b) shows that averaging across the token axis damages syntax similarity, lowering the blue curve across the whole network. Indeed, averaging over tokens ablates positional information, which is intuitively important for syntactic similarity. Nonetheless, the signal is still present, with similarity greater than $0.4$ at all depths.

Next, also in Figure 1, we look at what happens when we subtract from each sentence $\mathbf{X}_i$ its projections along its syntax centroid.[4] This procedure removes a significant fraction of the similarity between pairs for both concatenated (panel a)) and averaged (panel b)) representations. This suggests that syntax centroids capture important aspects of the syntactic information in the sentences, and that this information is linearly encoded given that it can be removed with a linear operation.

As a control, we misaligned the syntax centroids, that is, we subtracted the projection corresponding to a different, randomly-picked POS template from each sentence in a pair. We observe that this operation produces a smaller decrease in similarity in initial and final layers for concatenated representations, and in all layers for averaged representations, showing that our ablation targets specifically the syntax information carried by $\mathbf{X}_i$. We leave it to future work to determine why the ablation effect of mismatched centroids is particularly strong in the middle layers for concatenated representations.

### 3.2. Semantic similarity

Panels a) and b) of Fig. 2 show the similarity between a set of paraphrase pairs in English, represented by the concatenation or average of their last three tokens, respectively. Like

for syntax, the network is sensitive to semantic relatedness, displaying wide areas of high similarity. However, differently from what we observed for syntactically matched pairs, where similarity does not vary much as a function of depth, the similarity between paraphrases is low in the early layers of the network and high in the central ones. This is consistent with what was observed for the similarity between translations in different languages by Acevedo et al. (2025). A plausible interpretation is that early layers process the input, which, in the case of paraphrases, is characterized by different syntactic structures and partially different lexical material. Thus, initially the similarity between representations is low, since superficially the sentences are different. It then increases as a function of network depth, since the different words occurring in different syntax structures are combined and transformed by the network to create a meaning, which, by construction, is approximately the same for our paraphrases.[5] We could have expected late layers, being oriented towards output writing, to display lower similarity. This effect only clearly emerges at the very last layer, suggesting that the network is still carrying the semantic information built in the central layers until (almost) the very end of processing. Differently than for syntax, for semantics taking the average across the token axis (Fig. 2, panel b)) *increases* similarity with respect to concatenating the tokens (Fig. 2, panel a)). This is plausibly due to the fact that the paraphrases carry the same meaning using a different word order. Therefore, ablating positional information by averaging over the tokens helps highlight shared semantic contents. Finally, in Appendix E we show the similarity between the original English sentences and their translations in the other languages we use (another form of semantic matching). This observable behaves in a qualitatively similar manner to what reported in Fig. 2.

Next, we look at the effect of removing semantic centroids $\mathbf{T}_i$, constructed by averaging activations across translations. These vectors are our proxies to the "meaning" component contained in sentence representations. We observe that removing the projection of $\mathbf{X}_i$ along the direction of $\mathbf{T}_i$ for each sentence strongly reduces the similarity between paraphrases, and it does so predominantly in the central layers, where, as we just saw, semantics dominates similarity (green line in Fig. 2).[6] As a control, if we permute the semantic centroids so that each sentence is orthogonalized to an unrelated semantic centroid, the reduction in similarity between pairs is smaller (red line in Fig 2).

---

[3]As a general control on how meaningful the similarity values we report here and in the following experiments are, Fig. 7 of Appendix C shows that the similarity between randomly paired sentences is 0 at every layer.

[4]Fig. 8 of Appendix D shows an example where directly subtracting centroids from pairs of sentences (instead of their projections along them) can introduce spurious signals.

[5]As shown in Appendix J, in Gemma3-12b the "semantic phase" appears earlier, and it is followed by a dip in semantic similarity.

[6]In Appendix F, we show that the ablation effect would probably not become significantly stronger if we made semantic centroids more robust by adding more translations to them .

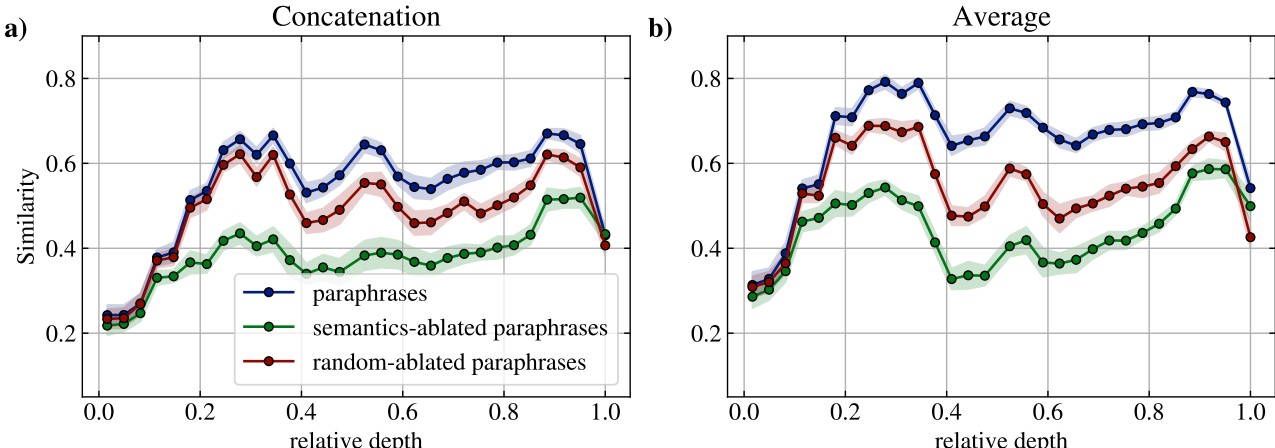

*Figure 2.* **Semantic similarity and its ablation**. Similarity between English sentences and their (English) paraphrases. Panels a) and b) represent sentences by token concatenation and average, respectively. The shaded colored areas represent 1 standard deviation, calculated by subsampling five times half of the samples.

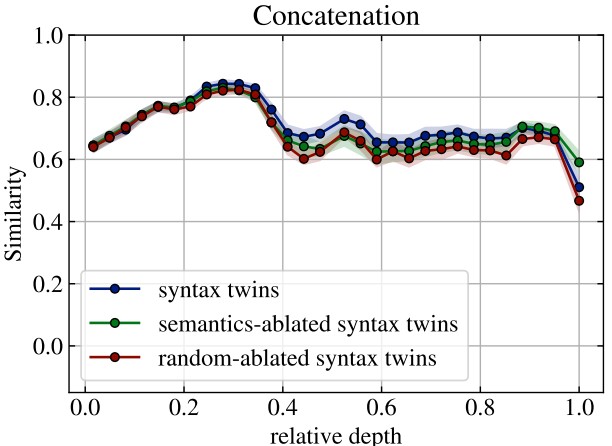

*Figure 3.* **Syntactic similarity with semantic ablation**. The shaded colored areas represent 1 standard deviation, calculated by subsampling five times half of the samples.

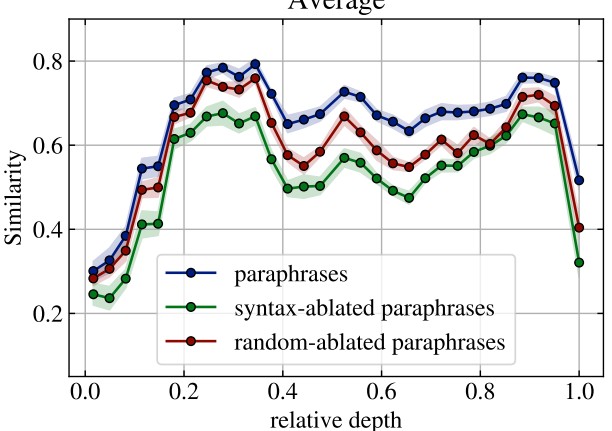

*Figure 4.* **Semantic similarity with syntax ablation**. The shaded colored areas represent 1 standard deviation, calculated by subsampling five times half of the samples.

### 3.3. Ablations across linguistic components

Having measured the syntactic and semantic similarity of hidden representations, we next ask if these two quantities can be dissociated. We start by removing the semantic centroids $\mathbf{T}_i$ (see Sec. 3.2) from syntactically matched pairs (see Sec. 3.1). Fig. 3 shows in blue, as a baseline, the similarities between the sentences $\mathbf{X}_i$ and their syntax twins $\mathbf{s}_i^0$, focusing on concatenated tokens since they have higher similarity scores than averaged tokens in Fig. 1.[7]

Remarkably, subtracting the projection of the activations

of each sentence and its syntax twin in the direction of the semantic centroid $\mathbf{T}_i$ largely preserves syntax similarity (green curve). As a control, if we subtract projections along the directions of randomly picked semantic centroids, that should be irrelevant to the target pairs, the effect is comparable (red line). These results thus suggest that semantics is encoded in a way that is approximately independent from syntactic information, so that syntax similarity is not significantly affected by the semantic ablation.

Reciprocally to the previous analysis, Fig. 4 shows semantic similarity between English paraphrases, as reported in Fig. 2, compared to the case in which we remove from each English sentence $\mathbf{X}_i$ its projection along its syntax centroid, $\mathbf{S}_i$, computed as in Sec. 3.1. We focus here on averaged tokens,

---

[7]The curve is almost identical to that in Fig. 1a), but computed on the last 3 tokens instead of 6 for compatibility with the semantic centroid: see Section 2.2.

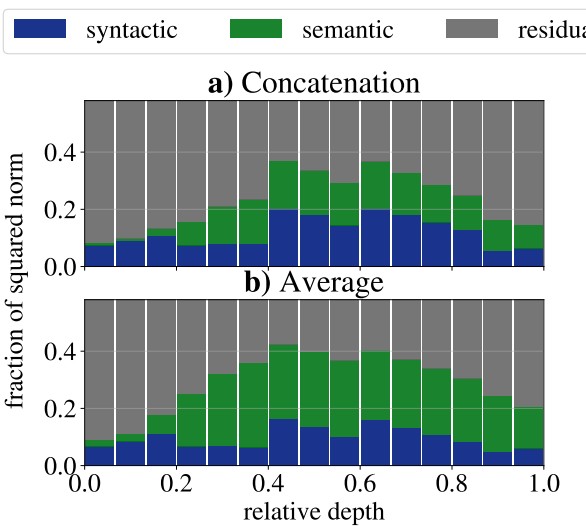

*Figure 5.* **Decomposition of sentence vectors.** Average fraction of squared norm from sentence activations $\boldsymbol{X}_i$ contained in syntax centroids $\boldsymbol{S}_i$ (blue) and semantic centroids $\boldsymbol{T}_i$ (green), across the network. The gray sections represent the residual fraction of norm that is not captured by either centroid. The vertical axis is cut to 0.6 for visualization purposes only.

where the strongest semantic signal emerged. The similarity between paraphrases when removing the syntactic centroid (green curve), while well above random levels, decreases significantly, especially in the central layers. This suggests that semantics is only partially separable from syntax. The red curve represents a control where the ablation is along syntax centroids picked at random from the dataset. In this case, the ablation effect is smaller than that produced by removing the correct syntax centroids.

An intuition for the asymmetry between Fig. 3 and Fig. 4 is the following. Removing semantic information from two syntactically matched pairs will not affect their syntax-based similarity, as the syntactic skeletons of the two sentences should remain intact. On the other hand, when stripping syntax, we might remove information (such as the position of each noun with respect to a verb) that might affect semantic similarity, leading to the observed asymmetry.

### 3.4. Decomposition of sentence vectors

Given that the syntax and semantic ablations have significant impact on representation similarities, it is natural to ask what proportion of $\mathbf{X}_i$ is given by its $\mathbf{S}_i$ and $\boldsymbol{T}_i$ components. In Fig. 5, we show the average fraction of square norm of sentence vectors $\boldsymbol{X}_i$ along the syntactic and semantic centroid directions. In gray, we show the fraction of the square norm that is not aligned with either, that we call "residual norm". For further details, see Appendix G. Consistently with our analyses above, we observe that the semantic component is larger in the averaged representation,

*Table 3.* **Effect of ablations on probes.** Best POS-template classification accuracy and paraphrase-recall@3 across all model layers. Bold numbers correspond to the minimum values of each column (strongest ablation) and underlined numbers correspond to maximum values of each column (best performance).

|  | Best syntax-acc | Best P-recall@3 |
|---|---|---|
| baseline (no ablation) | 0.85 | 0.85 |
| semantic ablation | 0.85 | **0.66** |
| syntactic ablation | **0.10** | 0.90 |
| semantic ablation (random) | 0.85 | 0.83 |
| syntactic ablation (random) | 0.81 | 0.85 |

whereas the syntax component is larger for concatenated tokens. Furthermore, the syntactic component is larger than the semantic one in the initial layers, whereas semantics dominates the central layers.

Note that our centroids account for a significant fraction of the total norm of the activation vectors, especially in the central layers, but nevertheless a big fraction of it is still not explained by them, a result we briefly come back to in the Conclusion. Finally, in Appendix H we computed the evolution of norm decomposition during training in the checkpoints made available for the Pythia-6.9b model. We found (Fig. 11) that the syntax centroids alone capture up to roughly half of the total squared norm very early in training, whereas the norm explained by the semantic centroids builds up progressively at later stages of training, again pointing at a separation between the encoding of syntax and semantics in LLMs.

### 3.5. Effect of ablations on probes

To get a sense of how our centroid ablations impact LLM behavior, we study their effect on two probes. As a syntax probe, we measure the accuracy of a linear classifier that, given a sentence $\mathbf{X}_i$, outputs its POS template. As a semantic probe, we consider a recall@3 metric: for each sentence $\boldsymbol{X}_i$, we compute its cosine similarity against all paraphrases $\mathbf{P}_i$, and we measure how often the correct paraphrase has one of the top-3 largest values.[8] For further details, see Appendix I. Table 3 shows the results of both probes, applied to the original sentence representations $\mathbf{X}_i$ (baseline performance), to the ablated sentences, and to sentences ablated through randomly permuted centroids, as a control. For simplicity, we only show for each row the maximum performance among all model layers.

In both experiments, we confirm that ablating the relevant

---

[8]Similar results were obtained with recall@1 (accuracy) and recall@2.

centroids (syntax centroids in the POS classification task and semantic centroids in the paraphrase recall task) has a much stronger effect than ablating permuted centroids (whose effect is minimal in both cases), or of ablating the task-irrelevant centroids (semantic centroids for POS classification and syntax centroids for paraphrase recall). Interestingly, the syntax ablation leads to a small *increase* (roughly 5%) in semantic recall.

## 4. Conclusion

We quantified syntactic and semantic similarities in LLM representations by analyzing local neighborhoods of sentence pairs sharing syntactic structure or meaning. Intuitively, syntactic similarity is already high in the initial layers, and remains roughly constant throughout the network. Furthermore, we found that, for each sentence represented by the vector $\mathbf{X}_i$, a significant fraction of the observed similarity is ablated by subtracting the projection of $\mathbf{X}_i$ along the direction of its syntax centroid, $\mathbf{S}_i$.

Semantic similarity is very low in the early layers, and it increases progressively with depth, being maximal in the inner layers of the network, and decreasing again on the last layer, which is oriented towards output generation. Again, subtracting the projection of each sentence $\mathbf{X}_i$ along the direction of its semantic centroid $\mathbf{T}_i$ strongly reduces semantic similarity, most clearly so in the inner layers.

We further studied to what extent syntactic and semantic information are coupled. Starting with syntax similarity, we found that removing from each sentence $\mathbf{X}_i$ its projection along the semantic centroid $\mathbf{T}_i$ does not significantly decrease its similarity to its syntax twin, $\mathbf{s}_i^0$. Instead, we found that the semantic similarity observed between the original sentences $\mathbf{X}_i$ and their paraphrases $\mathbf{P}_i$ is significantly affected when removing from $\mathbf{X}_i$ its projection along its syntax centroid $\mathbf{S}_i$, although the effect is not nearly as strong as that of ablating semantic centroids.

We next quantified the independent contributions of syntax and semantics centroids to sentence vectors across layers, confirming that the contribution of syntax is more stable across layers, whereas that of semantics concentrates in the inner layers. We found moreover that our syntactic and semantic centroids only account for a fraction of the information contained in sentence vectors. Future work should ascertain the extent to which this is due to our linear approach only capturing partial aspects of syntax and semantics (which is likely), and to what extent it stems from the fact that sentence vectors also encode a large amount of non-strictly linguistic information (along the lines of the "functional" knowledge explored by Mahowald et al., 2024).

Finally, we showed that our ablations also affect two downstream probes in the expected ways, and, in this case as well, it is possible to dissociate syntax and semantics.

The main takeaways are as follows. First, we confirmed with a new methodology that there is a "semantic core" in the way LLMs process linguistic data, concentrated in their inner layers. Second, an important portion of the syntactic and semantic information present in LLM sentence representations is encoded linearly, providing further evidence for linear coding as a general mechanism through which LLMs combine information. Third, syntactic and semantic information are to some degree separable, although it seems easier to independently capture syntactic information than the opposite. This is compatible with views from linguistics seeing syntax as an autonomous module of language. If this observation is confirmed in further studies, it could point to the autonomy of syntax as a universal property of linguistic representations, emerging in cognitive systems as different as human minds/brains and LLMs.

In future research, we want to combine our method to decouple syntax and semantics with a decomposition of neural activity into different time scales along the lines of Tamkin et al. (2020), under the hypothesis that syntax and semantics are predominantly encoded in high- and low-frequency bands, respectively. This would be consistent with our observation that syntax similarities are higher for concatenated tokens and semantic similarities for averaged representations, given that concatenated representations contain the highest frequencies (encoding changes in neural activity between nearby tokens), and the averaged representation itself is the lowest frequency mode (merging information from all tokens).

## Limitations

Our results were obtained with 4 separate LLMs, including the largest system whose weights are currently publicly available. Future work should systematically ascertain the extent to which the results depend on model size, amount of training data, and training objective. We should moreover explore what happens when focusing on languages different from English.

We generated our datasets through interfaces to currently state-of-the-art LLMs which have limitations and possible biases in their generation capabilities. This constrained our datasets to be of the order of about 2,000 samples and, maybe more importantly, each sentence to be of at most 10 words in length. We should scale to longer sentences, in order to see the dependence of our results on sentence length, and how that affects our aggregation methods.

Our similarity ablations in Figures 1 and 2 are only partially successful, leaving room for more effective ways to abstract syntactic and semantic information from sentence representations. Similarly, in Fig. 5 we explain at most roughly 40 %

of the squared norm of representations in central layers. It would be interesting to study how our results change using centroids constructed with more complex functions beyond representation averaging: for example, using non-linear feature extraction techniques such as those proposed by Wild et al. (2025).

We focused our analyses on representation similarity, and we did not perform interventional experiments during text generation. We expect our syntax and semantic centroids to be useful to steer the model away or towards specific meanings or syntactic structures, for example in relation to spurious correlations between syntax and semantics (Shaib et al., 2026), but we leave this application to future work.

We perform two probing experiments: one on linear syntax classification accuracy and one on semantic recall of paraphrases. Further tests along the lines of Hewitt & Liang (2019); Liu et al. (2019); Li & Subramani (2025) could provide deeper insights into how syntax and semantics are represented in LLMs, and they are left for future work.

## Code and Data Availability

Our code and dataset are publicly available at https://github.com/acevedo-s/syn-sem.

## Acknowledgments

We thank Valentino Maiorca and the ICML reviewers and area chair for useful feedback. We thank Arwa Osman, Emily Cheng, Thomas Brochhagen and Ecesu Ürker for linguistic support. MB received funding from the European Research Council (ERC) under the European Union's Horizon 2020 research and innovation program (grant agreement No. 101019291) and from the Catalan government (AGAUR grant SGR 2021 00470). AL and SA acknowledge financial support by the region Friuli Venezia Giulia (project F53C22001770002)

## Impact Statement

This paper presents work whose goal is to advance the field of Machine Learning. There are many potential societal consequences of our work, none which we feel must be specifically highlighted here.

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

# A. Datasets

## A.1. Shared syntax (equal POS structure)

Stimuli for the syntax experiments were generated using Gemini 2.5 Pro.[9]

We used the following prompt to obtain a set of candidate POS templates using Penn Treebank POS tags (Marcus et al., 1993):[10]

"Hi Gemini. Can you generate 250 different POS templates corresponding to well-formed English sentences, made of between 6 and 10 words, using the PennTreeBank tagset, and giving an example for each template, according to these rules:

- the templates correspond to well-formed English declarative sentences

- the sentences are made of a minimum of 6 words and a maximum of 10 words

- for each template, you produce an example sentence in word_pos format

- they should be all declarative sentences (no questions or imperatives)

- also, please avoid proper nouns"

The following prompt was then used to obtain sentences instantiating each template (the templates generated by the LLM were added at the end of the prompt):

"OK, thanks. Now please generate up to 100 sentences for each of the following templates.

Please follow these rules:

- the sentences are in format word_pos

- they must be varied within each template

- they must make sense in English

- if you can't come up with 100 sentences for a pattern, it is OK to generate less examples

- all sentences should be printed to a plain text file, with the set associated to each template preceded by the template

- please make sure the sentences are really perfectly matching the template"

Note that the LLM typically generated considerably less sentences than the requested 100. The output was manually checked to verify the naturalness of the generated sentences, with templates that were systematically populated by anomalous sentences being excluded.

Next, we filtered POS templates that have less than 5 sentences, in order to be able to compute syntactic centroids with at least 5 elements. This filtering procedure left us with 2,098 sentences belonging to 96 distinct POS templates, populated as shown in Fig. 6, left panel. The corresponding distribution of lengths is shown in the right panel of Fig. 6.

Finally, we constructed one syntax twin for each original sentence to be paired to it in the similarity computations. We used the prompt below, with Gemini (1/4 of the original sentences, in batches of 100) and ChatGPT4[11] (3/4 of the original sentences, in batches of 200), finding similar quality.

"Dear ChatGPT/Gemini, I will paste sets of lists that contain English sentences that instantiate various sequences of part of speech (POS).

The input lines have one sentence per line, in word_POS format, using the UPenn tagset.

For each input line, you should generate a single output line that contains two TAB-delimited fields: 1) the original sentence in word_POS format, and 2) a different sentence that has the same POS template, also in word_POS format.

Please do not generate more output than required.

---

[9]https://gemini.google.com/
[10]Penn Treebank POS list
[11]This was the latest version of ChatGPT at the time of this data collection.

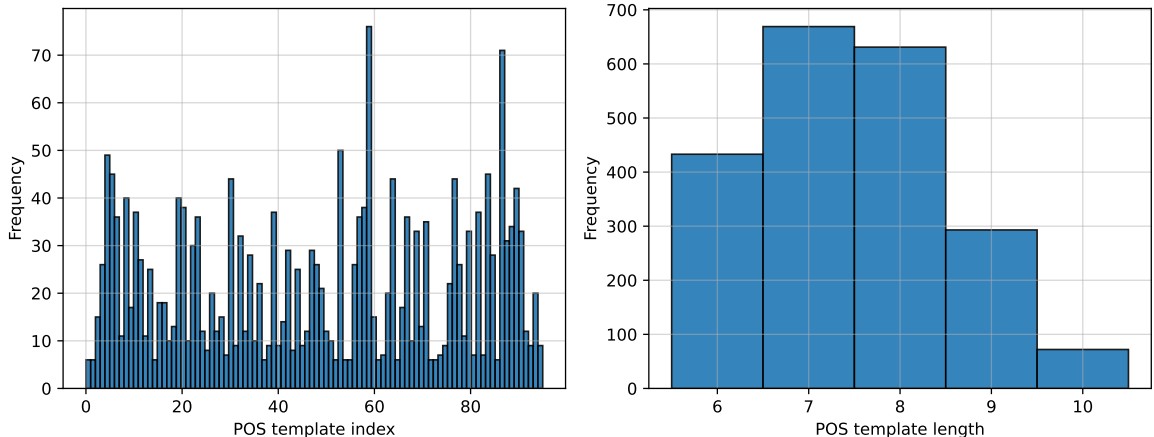

*Figure 6.* **Statistics of original sentences** $\mathbf{X}_i$ **after filtering.**

Please do not write a script: complete the task manually.

The generated sentences should be in grammatical, fluent and meaningful English, and very importantly, they should be very different from the input sentences in terms of both meaning and lexical items. Ideally they should be completely unrelated in meaning. Also, no generated sentence should be repeated across the batch.

Here comes a batch:"

The average word-overlap between original sentences and these syntax twins is $0.1 \pm 0.1$.

As also stressed in Section 2.1, the set of all original sentences $\mathbf{X}_i$ and the set of syntax twins $\mathbf{s}_i^0$ do not share any item. This allows to compute two sets of syntax centroids with no further shared information. We use syntax centroids constructed by original sentences to ablate syntax twins, and *vice versa*.

### A.2. Shared meaning I: Paraphrases

We generated paraphrases for the original sentences generated previously. To generate paraphrases, after experimenting with multiple LLMs and prompting styles, we found that the best strategy was to feed ChatGPT4 batches of 100 sentences at a time, using the following prompt:

" I am giving you a list of sentences in word_POS format. For each of these sentences, please generate a loose paraphrase with the following constraints:

- the two sentences should have different syntactic structures

- the paraphrase should sound natural in English

- paraphrasing should be achieved by a variety of means: main/relative clause changes, synonyms, prepositional phrases paraphrasing adverbs, passive/active, etc. Please use cleft constructions sparingly, only when they sound natural in English. The output file should have the original sentences in the same order, each followed by the paraphrase, also in word_POS format. The two sentences should be tab-delimited.

Thanks! "

Then, we filtered the data to have a word-overlap of less than 0.65 between a sentence and its paraphrase, leaving roughly 2,000 samples. These are the data used for Fig. 2 in the main text. In order to be able to remove syntactic centroids as in Fig. 4 of the main text, we discarded pairs of sentences in which the original sentence is not present in the final syntax-twin dataset, leaving roughly 1,600 sentence pairs.

**A.3. Shared meaning II: Translations**

We also translated all the original sentences into Arabic, Chinese, German, Italian, Spanish and Turkish. The languages were chosen based on a mixture of considerations of typological variety, support by the LLMs we are studying, and our ability to manually check translation quality. We found moreover that the best LLM selection and prompting strategy changed from language to language. For all languages, a native speaker or advanced L2 speaker checked at least 50 randomly selected translations. The process was iterated until all translations in the random sample were deemed acceptable, both in terms of faithfulness to the original and in terms of fluency.

For Spanish, German and Italian, we found that the best strategy was to use Gemini 2.5 Pro to translate as many sentences as afforded by our institutional subscription (about 2,100), and ChatGPT4 for the rest.

The Spanish/German/Italian Gemini prompt was:

" Dear Gemini, today I would like you to translate a set of sentences from English into Spanish/German/Italian. I will paste each sentence on a separate line. The translations should be plain and faithful, and sound natural in Spanish/German/Italian. The output should have tab-delimited lines, one for each line in the input, in the same order, with the English sentence followed by tab followed by the Spanish/German/Italian sentence. This should be just plain text, directly printed to your output window. Thanks! "

The ChatGPT4 prompt was:

" Dear ChatGPT, today I would like you to translate a set of sentences from English into Spanish/German/Italian. I will paste N batches of 200 English sentences, with a sentence on each separate line. The translations should be plain and faithful, and sound natural in Spanish/German/Italian. The output should have tab-delimited lines, one for each line in the input, in the same order, with the English sentence followed by tab followed by the Spanish/German/Italian sentence. This should be just plain text, directly printed to your output window. Please perform the task manually and without using Google Translate or other tools. I'll pass you each batch in turn. Thanks! "

For Chinese, we found in preliminary tests that DeepSeek3.1 provided the best translations. The prompt was:

" Dear DeepSeek, I would like you to translate English sentences into Chinese for me. I will upload a set of files, each containing 200 sentences. The output should be in plain text, with each English sentence in the same order as in the input, followed by a tab and the Chinese translation. The translations should be in a neutral tone, and sound natural. They don't need to be extremely faithful, if this affects how natural they sound. For example, the English sentences are often in the passive voice. It is OK to convert them to the active voice, if this helps fluency in Chinese. Thanks! "

By the time we started collecting sentences in Turkish and (standard) Arabic, ChatGPT5 became available to us, and we found in informal tests that it provided the best translations. The prompt was:

" Dear ChatGPT, today I would like you to translate a batch of sentences from English into Turkish/standard Arabic. I will pass you 14 text files, each with a list of English sentences, with a sentence on each separate line. The translations should be plain and faithful, and sound natural in Turkish/Arabic. The output should have tab-delimited lines, one for each line in the input, in the same order, with the English sentence followed by tab followed by the Turkish/standard Arabic sentence. This should be just plain text, directly printed to your output window. Please perform the task manually and without using Google Translate or other tools. I'll pass you each file in turn. Thanks! "

## B. Further Details on the Representation Similarity Measurements

To compute the similarity between two vector representations $A$ and $B$, one needs the distances between all pairs of data points in each feature space separately. For each sentence indexed by $i$, we thus have two sets of distances $d_{i,j}^A$ and $d_{i,j}^B$, $i, j = 1, ..., N_s$, respectively, where $N_s$ is the number of samples. Then, for each data point $i$, all other points $j$ are ranked in each feature space in increasing order, obtaining two sets of integer ranks $r_{i,j}^A$ and $r_{i,j}^B$. For example, $r_{ij}^A = 1$ and $r_{ik}^A = 2$ mean that $j$ is the first neighbor of $i$ in the representation space $A$, while data point $k$ is the second neighbor of $i$, respectively. Then, we compute the normalized average rank of points in space $B$ of the nearest neighbors in space $A$:

$$\frac{2}{N_s} \langle r^B | r^A = 1 \rangle = \frac{2}{N_s} \frac{1}{N_s} \sum_{i,j:r_{ij}^A = 1} r_{ij}^B$$

$$\equiv \Delta(A \to B),$$

(3)

where, we estimate the expected value over the dataset, $\langle . \rangle$, as the sample mean, and $\delta_{r_{ij}^A, 0}$ is a Kronecker delta fixing data samples $i, j$ to be first neighbors in space A. The normalization constant is chosen such that when representation $r^B$ is independent of $r^A$, and thus uniformly distributed between 1 and $N_s$, the quotient gives 1.

The quantity $\Delta(A \to B)$ was called Information Imbalance in Glielmo et al. (2022), and qualitatively, it is close to zero when close data neighbors in $A$ are also close neighbors in $B$, signaling a correlation between the feature spaces. Formally, the Information Imbalance is a measure that quantifies how much information about feature space B is carried by feature space A (Glielmo et al., 2022; Del Tatto et al., 2024), and it was used successfully in the context of deep learning representations by Cheng et al. (2025), who compared it to Central Kernel Aligment (Kornblith et al., 2019) (CKA), and by Acevedo et al. (2025), who compared it to both CKA and the Mutual k-Nearest Neighbor Alignment from Huh et al. (2024). Acevedo et al. (2025) and Huh et al. (2024) show that rank-based measures provide much stronger signal than CKA, and in particular Acevedo et al. (2025) shows that the Information Imbalance gives a signal comparable to that of Huh et al. (2024).

To turn the Information Imbalance into a symmetrical *similarity* measure, we compute 1 minus the average of $\Delta(A \to B)$ and $\Delta(B \to A)$. This is the similarity measure used for our analysis, see Eq. (2) of the main text.

We note that, if the main source of anisotropy in LLM representations is captured by a common bias or a shift vector, as discussed in Jørgensen et al. (2023) and Mu & Viswanath (2018), then our distance-based measure, unlike cosine similarity, is invariant by the shift since it corresponds to a global translation of all data points.

In order to remove possible outlier activations (Kovaleva et al., 2021; Bondarenko et al., 2023; Sun et al., 2024), we clip activations to quintiles of order $0.05$ and $0.95$ in all our experiments, following Huh et al. (2024); Acevedo et al. (2025). When measuring distances, we use normalized-$L_2$ distances, as Huh et al. (2024). Namely, we calculate regular $L_2$ distances between activations normalized to have unit norm. Note that the distances are calculated between the representations of three distinct and non-overlapping sets of sentences: The original sentences $\mathbf{X}_i$, the paraphrases $\mathbf{P}_i$, and the syntax twins $\mathbf{s}_i^0$, and then the neighborhoods of each index $i$ are compared across representations, for all $i$. The same procedure takes place for the ablated versions of each dataset.

Activations are taken from the residual stream, and in DeepSeek-V3, Qwen2-7b and Gemma13b have type bfloat16, which implies that distances have many ties. We break them at random by upgrading activations to float32 before computing distances.

## C. Random-Matching Control

Throughout the paper, we presented results for sentences that share syntactic or semantic information. Each original sentence $\mathbf{X}_i$, was paired with a syntax twin, $\mathbf{s}_i^0$, or a paraphrase, $\mathbf{P}_i$, and then their respective neighborhoods were compared. To show that the observed similarities are meaningful, in this section we misalign indices by doing a batch shuffle in the indices of the original sentences, thus breaking the syntactic or semantic correspondence. As a consequence, we compare local neighborhoods of unrelated samples. We see in Fig. 7 that the neighborhood structure of one set of sentences is not predictive of the other at all (similarity $\approx 0$), using for reference activations generated by DeepSeek-V3.

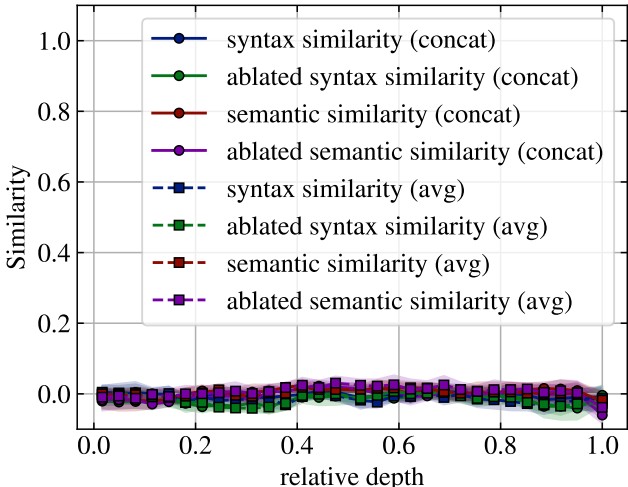

*Figure 7.* **Random matching control.** Syntactic and semantic similarities with and without ablation on misaligned data, i.e., performing a batch-shuffling on one of the spaces. This destroys similarities between representations. The shaded colored areas represent 1 standard deviation, calculated by subsampling five times half of the samples.

## D. Subtracting Syntax Centroids Instead of Projections

Fig. 8 shows in blue the similarity between English paraphrases processed by DeepSeek-V3 (same as Fig. 2). In green, it shows the similarity after subtracting the projection of $\mathbf{X}_i$ along the direction of the representation of a translation into Spanish, that we call $\mathbf{t}_i^{es}$. In purple, we show the similarity after the *subtraction* of the vector $\mathbf{t}_i^{es}$, instead of the removal of the *projection* of $\mathbf{X}_i$ along that direction. This introduces a strong similarity that we interpret as follows: When $\mathbf{X}_i$ has a negligible projection along $\mathbf{t}_i^{es}$, as in the initial layers of the network, removing it is irrelevant, and the similarity does not change much. But if this is the case, then the subtraction of the vector $\mathbf{t}_i^{es}$ from both the original sentence $\mathbf{X}_i$ and the paraphrase $\mathbf{P}_i$ introduces a spurious similarity between them, since they will start sharing an additional direction. We use here $\mathbf{t}_i^{es}$ instead of $\mathbf{T}_i$ for visualization purposes, since the effect is smaller, although still present, for $\mathbf{T}_i$.

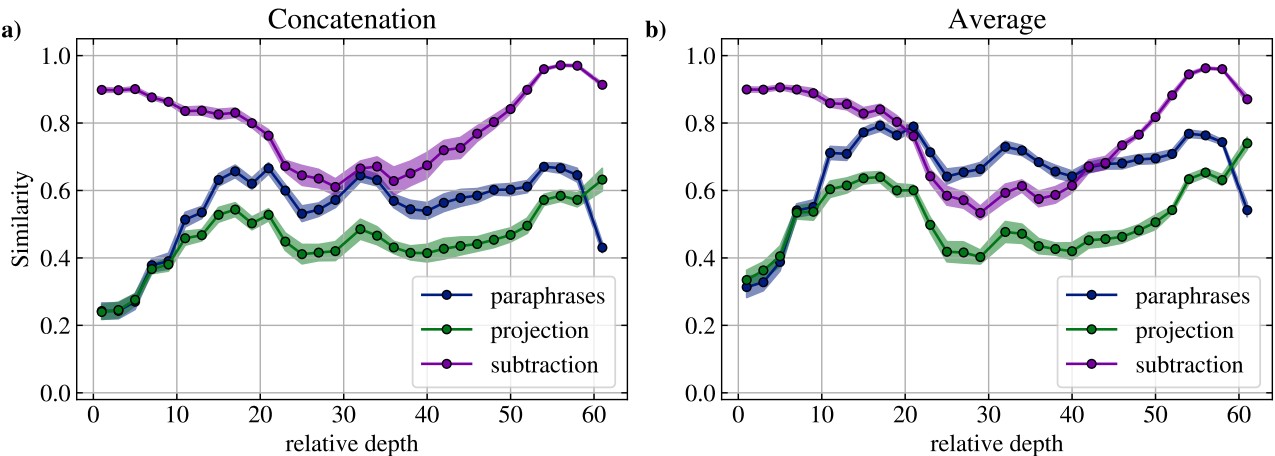

*Figure 8.* **Subtraction of centroids.** In blue, similarity between English paraphrases. In green, we remove the projection of $\mathbf{X}_i$ and $\mathbf{P}_i$ along the direction of the representation of a translation into Spanish, $\mathbf{t}_i^{es}$, similar to Eq. (1). In purple, we directly subtract from $\mathbf{X}_i$ and $\mathbf{P}_i$ the vector $\mathbf{t}_i^{es}$. The shaded colored areas represent 1 standard deviation, calculated by subsampling five times half of the samples.

## E. Similarity Between Translations

Fig. 9 shows the similarity between our original English sentences and their translations in each of the languages used to form the semantic centroids. The similarity profiles closely follow the one obtained for English paraphrases in Fig. 2, although with quantitative differences. We leave as material for future work the detailed study of these heterogeneities and why paraphrases display slightly smaller similarity values than translations.

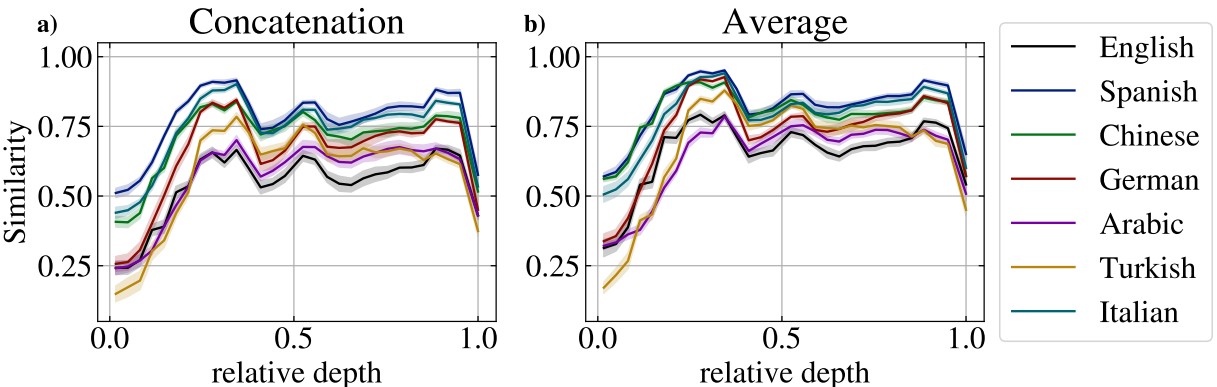

*Figure 9.* **Semantic similarities between translations.** Similarity between English sentences and all individual languages used in this work, including English paraphrases as a reference from the main text (cf. Fig. 2).

## F. Dependence of the Semantic Ablation on the Number of Languages in the Semantic Centroid

Fig. 10 shows the semantic ablation of paraphrase similarity given semantic centroids composed by pooling an increasing number of languages, up to the 6 we have data for. We don't find significant changes between 4 and 6 languages, suggesting that there is no need to include more languages, which is quite costly. Note that we averaged the results across different choices of language subsets, using cyclic permutations of the complete list, giving 6 possible options for any number of languages (except for the case of the whole list, which is unique).

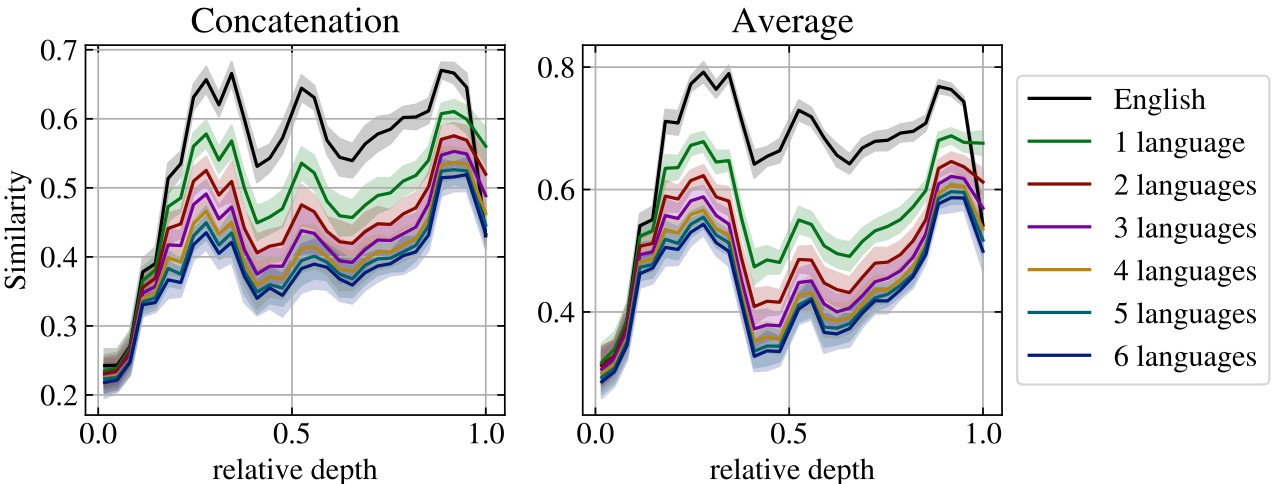

*Figure 10.* **Dependence of ablation on the number of languages entering the semantic centroids.** Similarity between English paraphrases with semantic ablations using semantic centroids $\mathbf{T}_i$ that contain from 1 up to 6 (all) of our available languages.

## G. Details on the Decomposition of Sentence Vectors

Given a vector $\mathbf{v} \in \mathbb{R}^E$, we can always decompose it into an orthogonal basis $\{\mathbf{b}_n\}_{n=1}^{E}$ with coefficients $c_n \in \mathbb{R}$ by

$$\mathbf{v} = \sum_{n=1}^{E} c_n \mathbf{b}_n, \tag{4}$$

where $\mathbf{b}_n \cdot \mathbf{b}_m = \delta_{nm}|\mathbf{b}_n|^2$, $\delta$ being the Kronecker delta, and $c_n = \mathbf{v} \cdot \mathbf{b}_n/|\mathbf{b}_n|^2$. Given decomposition (4), it follows that

$$\sum_{n=1}^{E} c_n^2 |\mathbf{b}_n|^2 = |\mathbf{v}|^2. \tag{5}$$

$E$ corresponds to the embedding dimension for averaged representations, or the embedding dimension multiplied by the number of tokens being used for concatenated representations. Following decomposition (5), in Fig. 5 the (normalized) average fraction of squared norm thus corresponds to the quantities

$$\begin{aligned}
\text{syntactic} &= \frac{1}{N_s} \sum_{i=1}^{N_s} \frac{(\mathbf{X}_i \cdot \mathbf{S}_i)^2}{|\boldsymbol{S}_i|^2 |\mathbf{X}_i|^2} \\
\text{semantic} &= \frac{1}{N_s} \sum_{i=1}^{N_s} \frac{(\mathbf{X}_i \cdot \mathbf{T}_i)^2}{|\boldsymbol{T}_i|^2 |\mathbf{X}_i|^2},
\end{aligned} \tag{6}$$

$N_s$ being the number of samples. Finally, the residual fraction in Fig. 5 is defined as $\text{residual} = 1 - \text{syntactic} - \text{semantic}$.

In practice, to reduce the anisotropy of representations, we remove from $\mathbf{X}_i$, $\mathbf{S}_i$ and $\mathbf{T}_i$ the global average vector of all data points $\mathbf{G} = \frac{1}{N_s} \sum_{i=1}^{N_s} \mathbf{X}_i$, similar to Mu & Viswanath, 2018 and Jørgensen et al. (2023). After this operation removes trivial alignments, $\mathbf{S}_i$ and $\mathbf{T}_i$ have on average a very small cosine similarity of $0.1 \pm 0.1$. In order to use the expansion in (5), we orthogonalize the centroids by removing from $\mathbf{S}_i$ its projection along $\mathbf{T}_i$, avoiding to count twice the overlapped norm.

## H. Norm Decomposition During Training

Fig. 11 shows the maximum squared norm across layers explained by the syntax and semantic centroids as a function of the training checkpoint of Pythia6.9b (Biderman et al., 2023), a model for which the training checkpoints are available.[12] The computations are done similarly to those of Fig. 5, and show that syntax norms peak much earlier during training than semantic norms, in agreement with our asymmetrical coupling discussed in relation to Fig. 3 and Fig. 4 from Sec. 3.3, where syntax similarity is roughly independent of semantics but the reverse is not true. Semantic centroids are orthogonalized with respect to the syntax centroids by removing the projection of the $\mathbf{T}_i$ vectors along the $\mathbf{S}_i$ vectors. Furthermore, Fig. 12 shows the layer profile of the norm decompositions evolving during training, where checkpoint 512 shows a massive increase in syntax norm with a different layer profile than the one observed at the end of training, whose understanding we leave as future research. Instead, semantic norms progressively build up in central layers at later stages of training.

## I. Details on the Probe Experiments

### I.1. POS-template classification

We performed multinomial logistic regression on the 96 different POS-templates, using Scikit-learn (Pedregosa et al., 2011). For this syntax experiment, we worked with concatenated representations of the last 3 tokens. The L2-regularization parameter $C$ (higher C corresponds to a weaker regularization) was set to $10^3$ after sweeping it between $10^{-3}$, $10^{-2}$, ..., $10^3$, $10^4$, $\inf$, tracking the best performance across layers. The training set corresponds to the set of syntax twins, $\mathbf{s}_i^0$, leaving the original sentences $\mathbf{X}_i$ and their different ablations as test sets, reported in Table 3.

As in Appendix G, we removed from all vectors their corresponding training or test global center $\mathbf{G} = \frac{1}{N_s} \sum_{i=1}^{N_s} \mathbf{X}_i$. To be consistent with the experiments on syntax and semantic similarities, where we computed the distances between vectors normalized to have unit norm, here we also normalize all vectors before probing.

---

[12]Pythia does not explicitly support the languages we use to construct our translation-based semantic centroids, but the fact that the results for the final checkpoint are quite aligned with those of the other models make us conjecture that the amount of non-English text leaked into its pre-training data is sufficient to provide a signal. This is in accordance with anecdotal evidence we collected elsewhere that this models does possess a degree of multilingual ability.

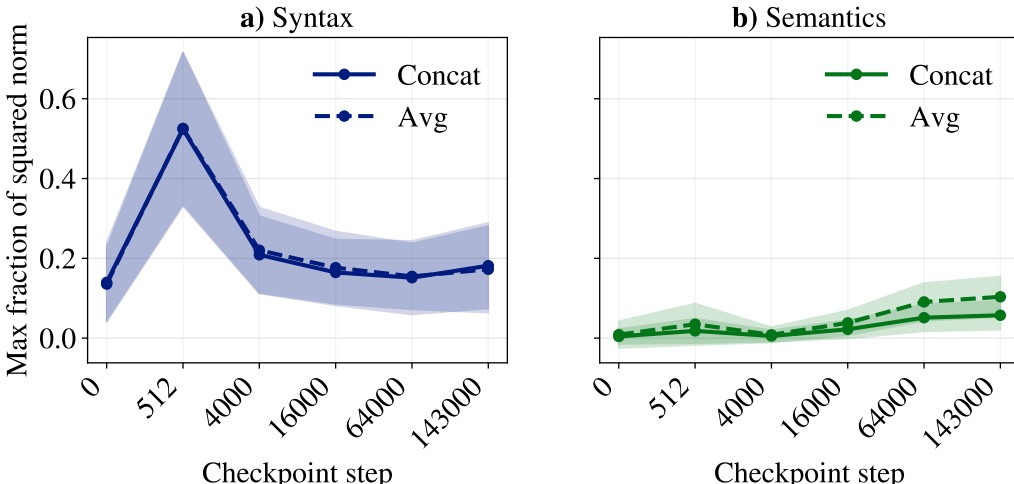

*Figure 11.* **Norm decompositions during training.** Maximum fraction of squared norm across the layers of Pythia 6.9b as a function of the training checkpoint. Checkpoint 0 corresponds to the untrained model, whereas checkpoint 143000 corresponds to the end of training. Panels a) and b) show the squared norm explained by the syntax and semantic centroids, respectively. Concat, Avg in the legend stand for concatenated and averaged representations, respectively, using the last 3 tokens of each sentence. The shadowed area corresponds to one standard deviation.

### I.2. Paraphrase-recall@3

For the paraphrase-recall experiments we used representations averaging the last 3 tokens of each sample. In order to compute cosine similarities, we mitigated the anisotropy of representations, similarly to Mu & Viswanath (2018) and Jørgensen et al. (2023), by removing from each original sentence $\mathbf{X}_i$, each syntax centroid $\mathbf{S}_i$, and each semantic centroid $\mathbf{T}_i$, the global average $\frac{1}{N_s}\sum_{i=1}^{N_s}\mathbf{X}_i$, and from each paraphrase $\mathbf{P}_i$, the global average $\frac{1}{N_s}\sum_{i=1}^{N_s}\mathbf{P}_i$. All results in Tables 3 and 4 are single-run experiments, so we have no variance estimation.

## J. More Language Models

In this section, we reproduce the main findings of our manuscript, obtained with representations from DeepSeek-V3, for two smaller LLMs: Qwen2-7b and Gemma3-12b. Fig. 13 shows that our syntax similarity results from Fig. 1 are remarkably robust across the three LLMs.

Figs. 14 and 15 show the semantic similarities computed for Qwen2-7b and Gemma3-12b, respectively, following the same procedure as for Fig. 2 in the main text. We observe that the qualitative behavior is comparable across models: there is a similarity maximum in inner layers that is significantly affected by the semantic ablation, but there are quantitative differences. In particular, the peak of semantic similarity of Gemma3-12b appears much earlier in the network with respect to the other two models, and is followed by a global minimum absent from either DeepSeek-V3 or Qwen2-7b.

Figures 16 and 17 show that Qwen2-7b and Gemma3-12b behave similarly to DeepSeek-V3 with respect to syntax similarity when ablating semantic centroids (cf. Fig. 3) and semantic similarity when ablating syntax centroids (cf. Fig. 4).

Figures 18 and 19 show that our decomposition of sentence vectors from Fig. 5 works similarly for Qwen2-7b and Gemma3-12b, respectively. Note that the semantic component (green) is larger for layers displaying higher similarity scores in Fig. 15.

Finally, Table 4 shows the results for POS-template classification accuracy and paraphrase-recall@3, on activations produced by Qwen2-7b (a) and Gemma3-12b (b), showing consistent behavior with DeepSeek-V3.

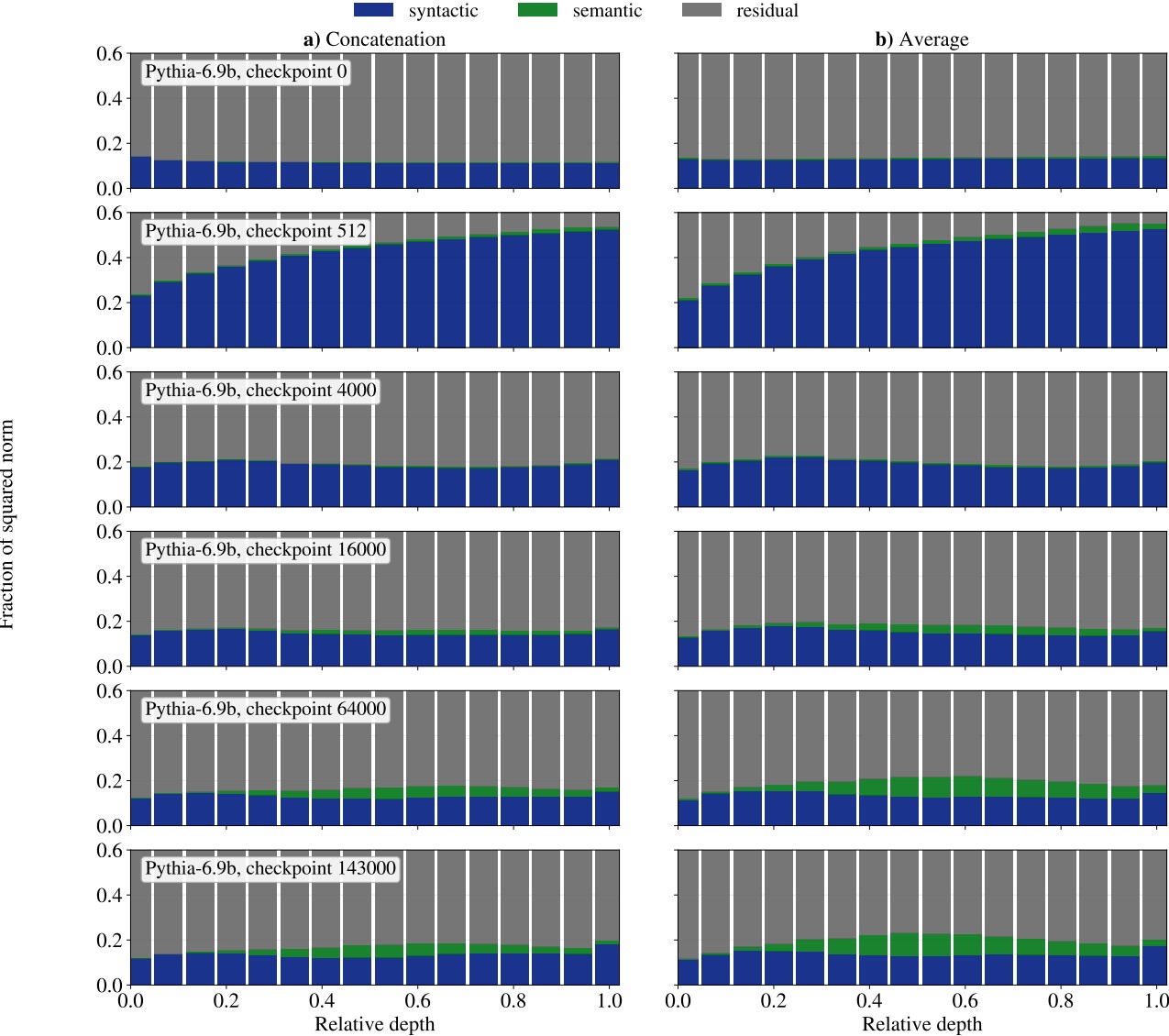

*Figure 12.* **Layer profile of norm decompositions during training.** Fraction of squared norm explained by syntax and semantic centroids using different training checkpoints of Pythia-6.9b for concatenated (panel a)) and averaged (panel b)) representations. Checkpoint 0 corresponds to the untrained model, whereas checkpoint 143000 corresponds to the end of training.

*Table 4.* **Effect of ablations on probes.** Best POS-template classification accuracy and paraphrase-recall@3 across all layers of (a) Qwen2-7b and (b) Gemma3-12b. Bold numbers correspond to the minimum values of each column (strongest ablation) and underlined numbers correspond to maximum values of each column (best performance).

| | Best syntax-acc | Best P-recall@3 |
|---|---|---|
| baseline (no ablation) | 0.82 | 0.91 |
| semantic ablation | 0.82 | **0.66** |
| syntactic ablation | **0.01** | 0.92 |
| semantic ablation (random) | 0.82 | 0.88 |
| syntactic ablation (random) | 0.79 | 0.90 |

*(a)* Qwen2-7b.

| | Best syntax-acc | Best P-recall@3 |
|---|---|---|
| baseline (no ablation) | 0.86 | 0.55 |
| semantic ablation | 0.86 | **0.42** |
| syntactic ablation | **0.01** | 0.64 |
| semantic ablation (random) | 0.85 | 0.53 |
| syntactic ablation (random) | 0.78 | 0.54 |

*(b)* Gemma3-12b.

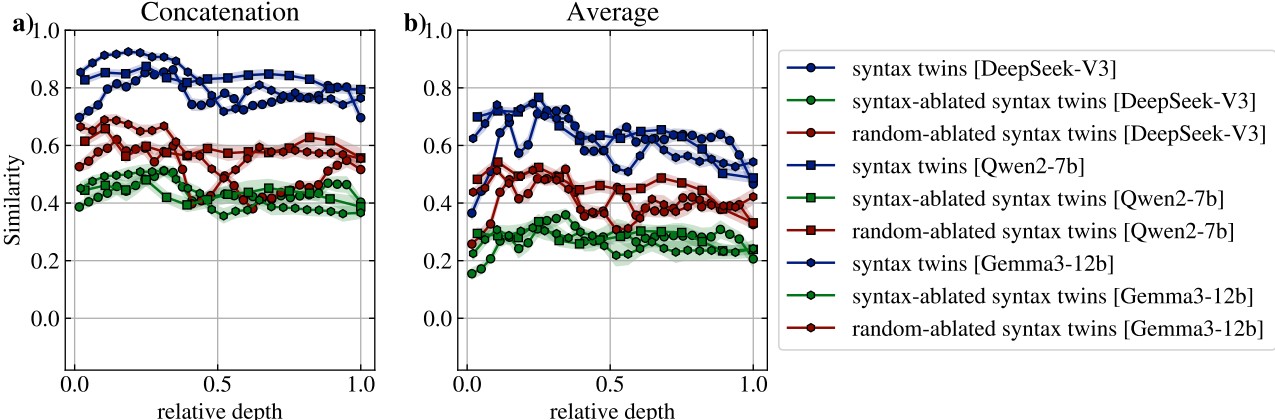

*Figure 13.* **Syntax similarity and its ablation across 3 LLMs**. Similarity between equal-syntax sentences (syntax twins), such as those presented in Table 1, for DeepSeek-V3 (same as in Fig 1), Qwen2-7b and Gemma3-12b. Panels a) and b) represent sentences by token concatenation and average, respectively. The shaded colored areas represent 1 standard deviation, calculated by subsampling five times half of the samples.

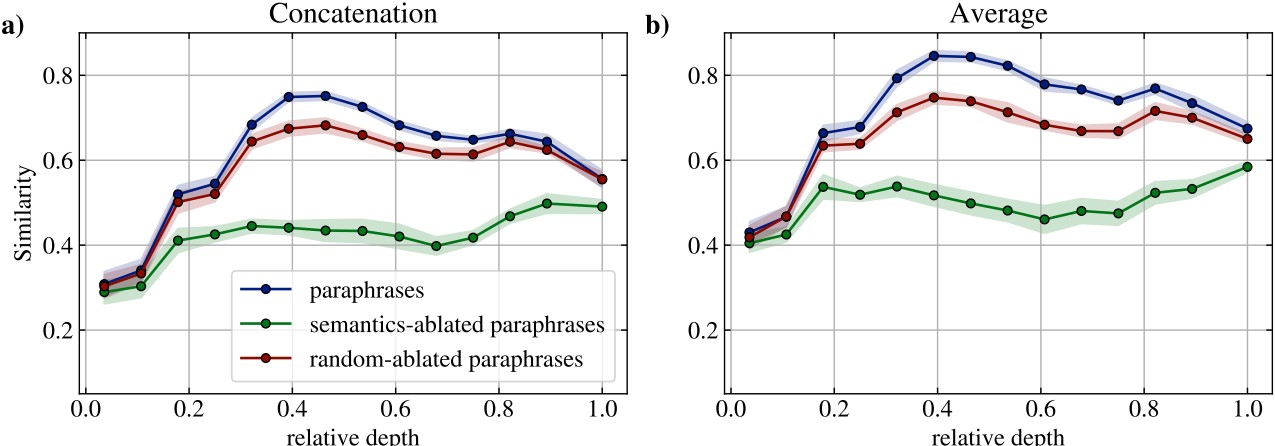

*Figure 14.* **Semantic similarity and its ablation for Qwen2-7b**. Similarity between English sentences and their (English) paraphrases, with the same setup of Fig. 2, using activations from Qwen2-7b. Panels a) and b) represent sentences by token concatenation and average, respectively. The shaded colored areas represent 1 standard deviation, calculated by subsampling five times half of the samples.

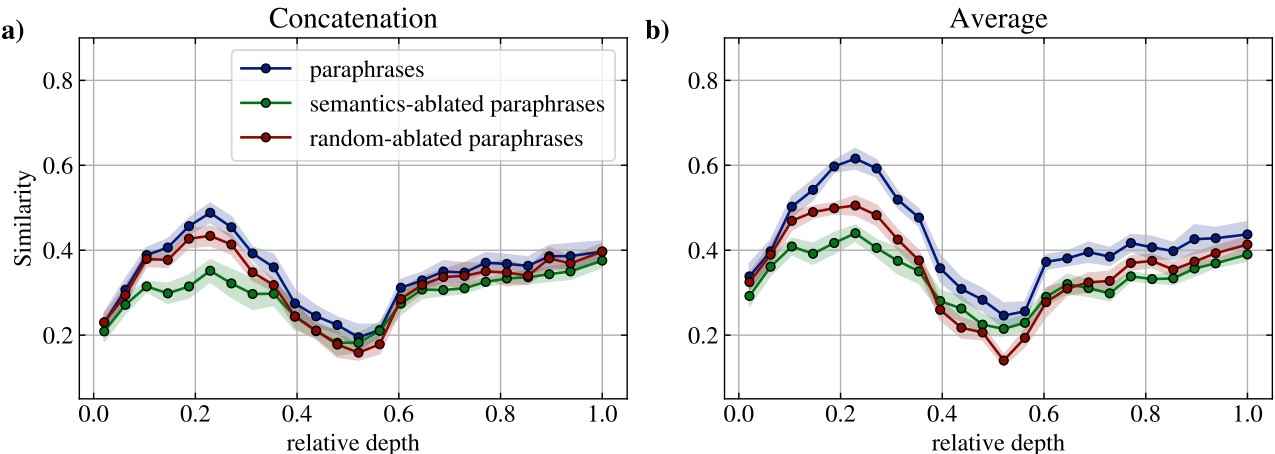

*Figure 15.* **Semantic similarity and its ablation for Gemma3-12b**. Similarity between English sentences and their (English) paraphrases, with the same setup of Fig. 2, using activations from Gemma3-12b. Panels a) and b) represent sentences by token concatenation and average, respectively. The shaded colored areas represent 1 standard deviation, calculated by subsampling five times half of the samples.

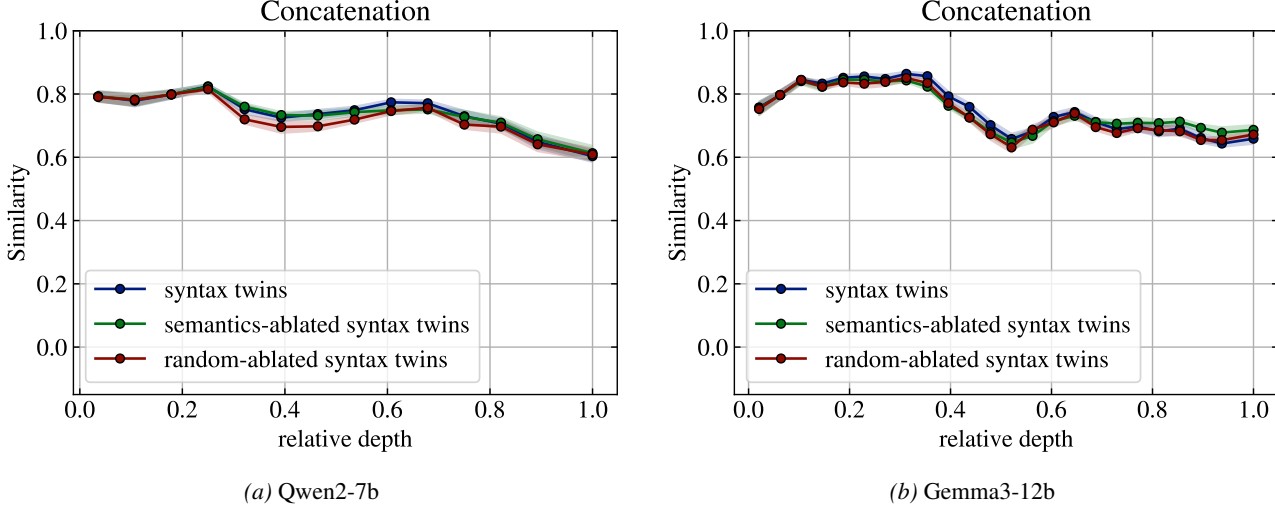

*(a)* Qwen2-7b          *(b)* Gemma3-12b

*Figure 16.* **Syntactic similarity with semantic ablation for Qwen2-7b and Gemma3-12b**. Same setup as Fig 3, on activations computed by (a) Qwen2-7b and (b) Gemma3-12b. The shaded colored areas represent 1 standard deviation, calculated by subsampling five times half of the samples.

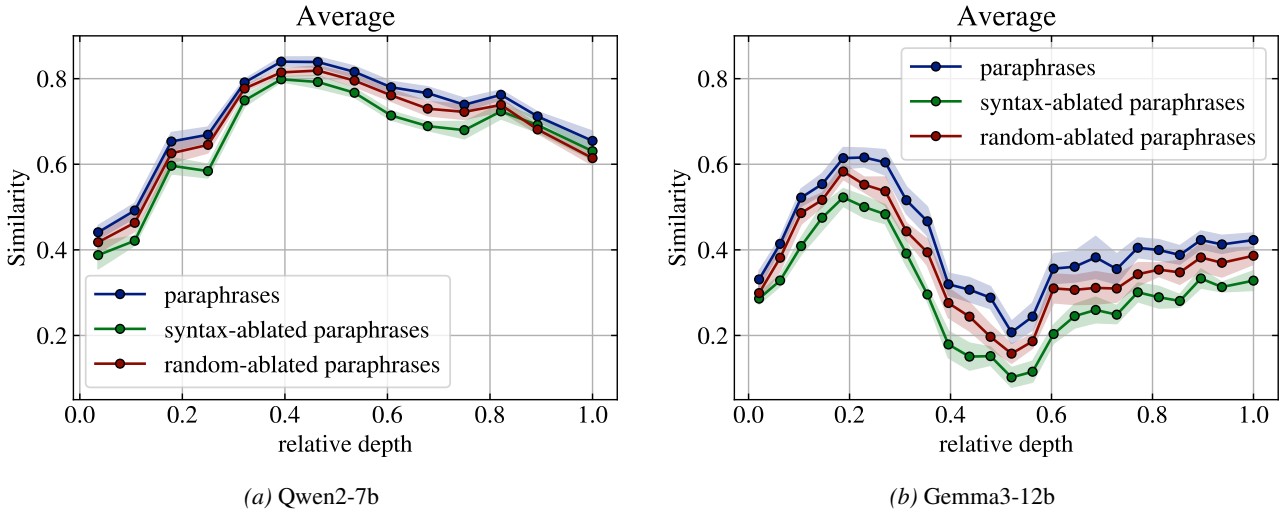

*Figure 17.* **Semantic similarity with syntax ablation for Qwen2-7b and Gemma3-12b**. Same setup as Fig 4, on activations computed by (a) Qwen2-7b and (b) Gemma3-12b. The shaded colored areas represent 1 standard deviation, calculated by subsampling five times half of the samples.

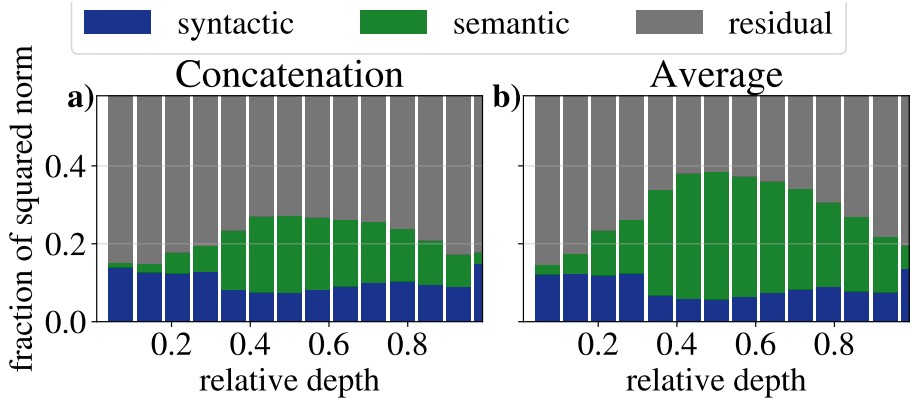

*Figure 18.* **Decomposition of sentence vectors for Qwen2-7b**. Same setup as Fig 5, on activations computed by Qwen2-7b.

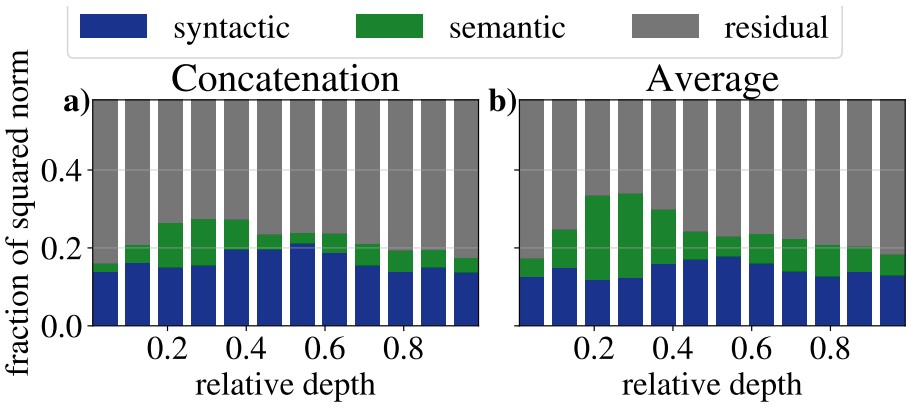

*Figure 19.* **Decomposition of sentence vectors for Gemma3-12b**. Same setup as Fig 5, on activations computed by Gemma3-12b.

# K. Assets

We run DeepSeek-V3 on a cluster of 16 H100 GPUs (80GB each), using the SGLang framework (Zheng et al., 2024). Both Qwen2-7b and Gemma3-12b fit a single H100 GPU. The former was also run within SGLang, and the latter through Huggingface.[13] Each loading of representations, computation of a single similarity curve, probe or norm decomposition takes roughly between one and two minutes. The computation of distances between representations is performed in the GPU with ad-hoc code implemented in JAX.

## K.1. Licenses

DeepSeek-V3: https://huggingface.co/deepseek-ai/DeepSeek-V3 ; Model licence.

Qwen2-7b: https://huggingface.co/Qwen/Qwen2-7B ; model license: Apache-2.0

Gemma3-12b: https://huggingface.co/google/gemma-3-12b-pt ; model license: Gemma

Pythia-6.9b: https://huggingface.co/EleutherAI/pythia-6.9b ; model license: Apache 2.0.

Scikit-learn license: BSD 3-Clause ("New/Modified BSD").

---

[13]https://huggingface.co/

