# OpenReview forum: "Differential syntactic and semantic encoding in LLMs"
_ICML.cc/2026/Conference — ICML 2026 regular_

### Official Review · Reviewer_sypB · 2026-03-03

**Soundness:** 3
**Presentation:** 3
**Significance:** 2
**Originality:** 2
**Overall Recommendation:** 5
**Confidence:** 4

**Summary:**

The paper addresses a key challenge in linguistics, understanding the code and representational overlap between syntax and semantics in natural language. To do that, inner representations of a Large Language Model are investigated. The paper relies on a simple approach, first computing an embedding per sentence by averaging (or concatenating) across tokens, and then averaging these sentence embeddings across sentences. This procedure is done for sentences drawn from the same syntactic template or with the same semantics, via 6 additional languages. The resulting centroids summarize how a given syntactic structure or meaning are represented in the LLMs hidden states.

Given such centroids, the semantic and syntactic components are projected out (ablated) of the sentence embeddings. This allows a comparison of the similarity of sentence embeddings across layers (1) without ablation (2) with a (semantic, syntactic or random) ablation.

The results show that ablating syntax hurts the similarity of syntactic twins more than both a random ablation and not ablating at all. The same result applies to semantics, where ablating the centroid consistently hurts similarity. The paper also compares concatenation versus averaging, showing that averaging has a negative impact on similarity for syntax but not for semantics. Furthermore ablating semantics is shown not to affect syntactic similarity but ablating syntax does have an effect of semantic similarity, resulting in an intriguing asymmetry. Then, the authors compute the contribution of both syntax and semantics to the layer representations of the LLM, showing that semantics mostly appears at middle layers and syntax at early ones. Finally, probes are trained to predict the syntactic template of each sentence from their sentence representations. For semantics, a retrieval task is considered. Ablating each component reliably modulates classification performance for syntax and semantics independently.

**Compliance With Llm Reviewing Policy:**

Affirmed.

**Final Justification:**

The rebuttal addressed my points.

**Key Questions For Authors:**

- POS templates are used to generate the stimuli. The paper mentions that given a template, there can be sentences with different parses. I wonder whether this could be further investigated. For example, having a dataset with sentences drawn from the same template but admitting different parses, and seeing whether the syntactic similarity forms a cluster for each of the parses.
- What happens as a function of pre-training? Can these results be extended to the Olmo or Pythia family?
- What do the results look like when considering only the last token representation of the sentence instead of concatenating a few?

**Limitations:**

yes

**Strengths And Weaknesses:**

- While pooling is well-motivated and keeps the message of the paper simple, I find the choices (averaging vs. concatenating the last N tokens) arbitrary. I am skeptical about concatenating the last N tokens for the case of syntax since this could include a lot of information about the template, artificially inflating the similarity score. For example: given 2 templates shown as example in the paper (PRP VBD RB VBN IN DT NN and PRP VBD DT JJ NN RB) these 2 can be easily differentiated if the token representations of the last 3 tokens at that given layer reflect the POS tag (IN DT NN vs. JJ NN RB). Therefore, I am not convinced that concatenation vs. averaging can be compared.
- More generally, for the concatenation setup, it could be that the centroid reflects a sequence of POS, in that case, it is not surprising that the representation is linear. Also, if that is the case, I think further experiments should be done to better understand the subtle difference between having distinct POS sequences and syntactic parses.

---

> ### Author Rebuttal · Authors · 2026-03-30
>
> Thanks for acknowledging the importance of our research and the strength of our results.
>
> Pooling methods and syntax:
>
> We agree that concatenation pooling is injecting some prior knowledge that might favor the extraction of syntactic information, by giving more weight to same-token comparisons (modulo cross-token information spreading through attention). Still, while the presence of purely syntactic information might be more easily detected with this representation, 1) syntactic information (e.g., information about POS sequences) must still be encoded in the representation, or we could not pick it up; 2) importantly, all our results are also confirmed with average pooling (and now last-token representations, see next), suggesting they are not an artefact of concatenated representations.
>
> We have now repeated the experiments with last-token representations, again replicating all our main trends, with noisier quantitative results. We will add these experiments to an appendix.
>
> The idea of looking at different syntactic trees corresponding to the same POS template is interesting, although not easy to implement, in particular because of the presence of ambiguities between the possible parses. We are however exploring this as a direction for future work, and we will mention this explicitly in the discussion.
>
> Emergence of syntax and semantic during pre-training:
>
> Thanks for suggesting this. We have now repeated our analysis on 5 Pythia-6.9b training checkpoints. Syntax appears to emerge first: very early in training, the syntax fraction of the squared norm averaged across layers has a peak, reaching 0.4 of the total norm. The corresponding semantics value is only 0.04. Later, the syntax fraction decreases, whereas the semantic component strongly increases, both converging to comparable values around 0.1. These preliminary results are aligned with our main findings, of syntax and semantics being encoded differentially, with asymmetric dependencies between them, and moreover show an intriguing asymmetry in learning. We will add them in an appendix.

---

> > ### Author Rebuttal · Reviewer_sypB · 2026-04-03
> >
> > Thanks to the authors, all my points were addressed. I have updated the score accordingly.

---

> > > ### Author Response · Authors · 2026-04-06
> > >
> > > We thank the referee again for their constructive feedback and their positive evaluation of our manuscript.

---

### Official Review · Reviewer_Ae9Q · 2026-03-08

**Soundness:** 2
**Presentation:** 2
**Significance:** 2
**Originality:** 2
**Overall Recommendation:** 4
**Confidence:** 4

**Summary:**

The authors study how syntactic and semantic information is encoded in LLMs, with a case study on DeepSeek-V3. They run experiments ablating syntactic and semantic information from both syntactic and semantic clusters (that are concatenated from the last N tokens or averaged). In their setup, they create syntactic twins and multiple semantic paraphrases. For the syntax twins, they see whether syntax can be ablated from the centroid representation; likewise, for the paraphrases, they see whether semantics can be ablated from the centroid representation. Finally, they find that syntax and semantics are disassociated by subtracting the direction of the syntax twin from the semantics centroid, and the semantics paraphrases from the syntax centroid. In the case where they subtract the direction of the syntax twin from the semantics centroid, similarity is not preserved. In the case where they subtract the direction of the paraphrases from the syntax centroid, similarity is. They also perform similar ablations on probes.

**Compliance With Llm Reviewing Policy:**

Affirmed.

**Final Justification:**

Authors addressed the changes I mentioned with new experiments to be added to the main body and appendix and am satisfied with it.

**Key Questions For Authors:**

1. Why can we assume that translations into different languages will impart similar semantic meaning? I found a paper [1] that could be evidence, but I would like more elaboration (especially in the paper).
2. Why were 6 tokens for syntax and 3 for semantics chosen? What if you ablate along the number of tokens?
3. I do not understand the intuition about why one would want to concatenate tokens to measure syntax and semantics. Is there a particular reason?
4. Why is the syntactic similarity with semantic ablation shown with concatenation, and the semantic similarity with syntax ablation shown with averaging? Shouldn't both be shown in the main body?
5. What is the setup for the syntax probe? How does one output a single POS template for a sentence $\mathbf{X}_\text{i}$, given that it can contain many words in one sentence?
6. How was the recall@3 metric chosen? Why 3?
7. Lines 414-419, namely this part of the paragraph: "...given that concatenated representations contain the highest frequencies, and the averaged representation itself is the lowest frequency mode." This seems a bit random, where was it shown that concatenated representations = highest frequencies, and averaged representations = lowest frequencies?

[1] Chang et al. 2022. The Geometry of Multilingual Language Model Representations.

**Limitations:**

yes/somewhat -- though my main complaint is for instance in the limitations, the authors say: "Future work should systematically ascertain the extent to which the results depend on model size, amount of training data, and training objective. We should moreover explore what happens when focusing on languages different from English." This seems like the entire investigative point of what machine learning research is -- to see how a task might scale with model size, training data, and objective. I'd like some more specific limitation for this part, since it's too general.

**Strengths And Weaknesses:**

**Strengths**:
1. The authors present a simple experiment to characterize syntax vs. semantics, which is simply mean averaging (although some of these approaches have previously been explored in prior work).
2. I find that the findings are cool -- that syntax centroids are not affected by semantics, but semantics is affected by syntax. Intuitively, I do think this somewhat makes sense, given that there are many ways to state a semantically-similar sentence that may not be syntactically similar, but changing the semantics may result in different templated syntactic behavior.

**Weaknesses**:
1. In the related work, the authors seem to pose that many prior works often don't investigate an aspect of syntax and semantics, which is: "However, what kind of linguistic processing [this shared process] is remains an open question." I feel like the work does not adequately address this question (given that this is stated in contrast to these works). "Kind of linguistic processing" is a bit vague to begin with, so what does this work offer that other contemporary works might not offer? It would be a good idea to elaborate in the same related works section what sets your contributions apart from other works that investigate the same theme, because it's currently unclear.
2. There are some related works that I think could be cited that aren't. For instance: [1] and [2]. For [1], it's been shown that models can succeed with syntactic mastery but fail on semantics. For [2], I think the conceptualization of "syntactic templates" (and thus syntax twins) falls back to a similar idea from this paper, where models may be sensitive to templated knowledge in pre-training and thus rely more on templated knowledge rather than semantic information. The findings from the authors' work fall in line pretty nicely with the two additional prior works. Additionally, these findings and [2] also imply that understanding how these syntax and semantics centroids work across training may be fairly important. For instance, an interesting direction could be to ask what emerges first, whether it is syntax or semantics, since their findings seem to imply that one might emerge before another.
3. The paper as written leaves the reader confused about the contributions. In each section, the authors offer particular ideas for future work, so we never get a set of strong, concrete takeaways. I highly recommend relegating these "future work" contributions to the end of the paper, rather than interspersing them throughout the paper.
4. In section 3.2 and 3.3 it's fairly repetitive; in 3.2 there are two paragraphs, but the authors talk about the same things in both paragraphs. I'd remove the repetition. Also, I'd recommend introducing the random baseline earlier because, when referencing the figures, the reader is left confused about the definition of this "random" baseline.
5. Authors claim to experiment with multiple models, but these results should be in the main body (since the authors don't use the full 8 pages, it leaves the reader questioning why extra space wasn't filled with these additional results).
6. I am concerned that the last N representation will be overly biased towards sentences that are shorter. So as the sentence lengths vary significantly, the results may not hold, since activations from shorter sentences may contain complete information that longer sentences may lose.
7. I think that norm averaging being larger for semantics, norm concatenation being larger for syntax, makes sense intuitively. For instance, if syntactic pairs are more compressible (because for instance, [2] shows that models rely on syntactic templates during pre-training, which might mean the representation is more compact), then the only way to obtain a high norm is to take the norm over the concatenated activations, not the averaged activations. On the other hand, semantic paraphrases might mean the same thing but be represented wildly different, which may account for a higher norm when averaging the activation dimensions. I guess all I am saying here is, I don't understand what I should take away from this section -- what is the implication of one method obtaining a higher norm than the other, because this could be a result of how the activations are aggregated?
8. To fully claim that semantics and syntax is disassociated, I would be convinced by having a causal experiment that tests whether changing syntactic structure while holding semantics constant might result in differing semantic performance. It's possible that the existing setup may be picking up on features that are synonymous with syntax or semantics, but not directly causal. Currently, there isn't a causal experiment that the authors have authored up, and although it is acknowledged in the limitations, many interpretability studies generally confirm their findings with probing *and* causal experiments (like [3], [4]).
9. Some language needs to be sharpened (few run on sentence, wrong words used). Similar to pt4, some of the sections are unclear and repetitive and sentences are run-ons. For ex (did not exhaustively look through all issues):
- Line 255 -> This observable behaves in a qualitatively similar manner to what reported in Fig. 2. "observable" --> observation
- Line 256 -> for semantics taking the average across the token axis (Fig. 2, panel b)) --> missing a comma and reads like a run-on
- Line 250 -> Therefore, ablating positional information by averaging over the tokens helps highlighting shared semantic --> "highlighting" -> "highlight"

[1] Mahowald et al. 2023. Dissociating language and thought in large language models.

[2] Shaib et al. 2024. Detection and Measurement of Syntactic Templates in Generated Text.

[3] Marks et al. 2023. The Geometry of Truth: Emergent Linear Structure in Large Language Model Representations of True/False Datasets.

[4] Elazar et al. 2020. Amnesic Probing: Behavioral Explanation with Amnesic Counterfactuals.

---

> ### Author Rebuttal · Authors · 2026-03-30
>
> Thanks for recognizing the excitement of our results and for the suggestions to improve their presentation. We will incorporate them in our revision.
>
> Contributions with respect to the literature
>
> We are not aware of any work showing, by a quantitative analysis, that linear representations of abstract semantics and syntax can be asymmetrically disentangled, and quantifying their independent contributions across layers. Thanks for the further relevant references, that we will cite in our revision.
>
> Emergence of syntax and semantics during training
>
> We have obtained preliminary results on 5 training checkpoints from Pythia-6.9b. Syntax appears indeed to emerge first: very early in training, the syntax fraction of the squared norm averaged across layers has a peak, reaching 0.4 of the total norm. The corresponding semantics value is only 0.04. Later, the syntax fraction decreases, whereas the semantic component strongly increases, both converging to values around 0.1.  These preliminary results are aligned with our main findings, of syntax and semantics being encoded differentially, with asymmetric dependencies between them. We will add them in an appendix.
>
> Representation methods
>
> We share the interpretation given by the reviewer in Weakness 7, that is aligned with our Figures 1 and 2, showing that concatenated and averaged representations give stronger signals for syntax and semantic similarity, respectively. We present our Figure 5 as yet another way to observe the same phenomenon, where the amount of explained norm by our centroids is a proxy of the amount of information captured by each representation.
> As asked by another reviewer, we reproduced our findings with last-token-only representations, which behaved qualitatively equal to the other two aggregation methods, but providing weaker signals for syntax. This shows that our results are not overly dependent on representation choice.
> As stated in our limitations, the length dependence of our results is an important future line of work, complicated by the representational nuances arising in longer sentences and the fact that they tend to feature more parsing ambiguities.
>
> Causal experiments
>
> We agree with you, and we were careful to avoid claims about causality or complete independence of syntax and semantics. Our main findings are centered on neighborhood structures, supported by the norm decompositions and probes. Also, our ablation experiments are interventional, showing that removing the syntactic component affects a downstream syntactic task, and removing the semantic component affects a semantic task, whereas syntactic ablation does not affect the semantic task and, conversely, semantic ablation does not affect the syntactic task (Table 3). It would be extremely interesting to use our centroids, for example, to steer a sentence representation from active to passive without changing its meaning, but due to the complexity and scope of such experiments, they are left for future work, as stated in the limitations.
>
> Q1: Can you kindly clarify what you mean by “imparting”? We think that the fact that translations (of simple sentences) have similar meanings is not controversial.
>
> Q2: As stated at the end of Sec. 2.1, we used as many tokens as possible for each dataset. The maximum number of tokens that one can concatenate is given by the minimum token length across sentences in a dataset. Importantly, we now performed dedicated tests in which the analysis is performed only on sentences with at least 6 tokens, and we observed no significant differences. We will add this analysis in the appendix.
>
> Q3: We always tested the signal strength of the results with both methods, and found that concatenation provides stronger signals for syntax, whereas averaging provides stronger signals for semantics.
>
> Q4: As stated in Sec 3.3, we simply chose the representation that gave the strongest signal in Figs 1 and 2. We checked that the results from Figs 3 and 4 are similar when using both aggregation methods. We will try to fit all results in the main text of the revision.
>
> Q5: We treat whole POS templates as separate classes to be predicted by the classifier, not as composite sequences. Please see Appendix H1 for a detailed explanation.
>
> Q6: Recall@3 was an arbitrary choice. Recall@2 and Recall@1 behave similarly, and will be added.
>
> Q7: If one takes the Fourier transform of a neuron x along the token axis with
> y (k) ~  \sum_t x(t) exp( i k x ), and sets k = 0,
> then y(k=0) is proportional to the average of the signal across token positions, t (standard result in signal processing and physics). Concatenating tokens along the token axis explicitly retains positional information, thus allowing to compute the different y(k)’s, for all available k’s.

---

> > ### Author Rebuttal · Reviewer_Ae9Q · 2026-04-03
> >
> > Thank you to the authors for comprehensively answering and adding experiments for what I've requested. I think the findings are interesting for scientific reasons. In response to Q1: what I am talking about is to include a citation that the representations between different languages would be the same. It might not be obvious to those who are not in the space of interpretability.

---

> > > ### Author Response · Authors · 2026-04-06
> > >
> > > We thank the referee for their positive evaluation on our manuscript. We’ll make sure that the camera-ready version of our manuscript will be clearer about Q1, including references, and the rest of the previous questions and concerns.

---

### Official Review · Reviewer_9xMQ · 2026-03-13

**Soundness:** 4
**Presentation:** 4
**Significance:** 4
**Originality:** 4
**Overall Recommendation:** 6
**Confidence:** 4

**Summary:**

This paper investigates the extent to which syntax and semantics can be disentangled in the internal representations of LLMs. To do so, it introduces a new way to produce syntax-only and semantics-only representations: for a given syntactic structure, the representations of many sentences with the same syntax can be averaged to give a “syntactic centroid” that encodes the syntax of any sentence with that syntax, and similarly for semantics (using translations of the sentence to multiple languages to create sentences with the same meaning). The authors verify in various ways that these centroids indeed capture syntax/semantics: removing (via a projection and subtraction) the syntactic centroid can decrease the extent to which a sentence is measured as similar to its syntactic neighbors (but not its semantic ones), and vice versa for ablating the semantic centroid. Further, when syntax is ablated, probes trained to extract syntax drop in performance while probes trained to extract semantics show little effect - and vice versa for ablating semantics. These results show that these centroids are effective encodings of syntax and semantics, further demonstrating that syntax and semantics are encoded in largely disentanglable ways and that both are encoded at least approximately linearly.

**Compliance With Llm Reviewing Policy:**

Affirmed.

**Final Justification:**

The rebuttal reinforced my strong view of the paper, so I am retaining my high score. Because I assessed the paper as being strong along all dimensions of soundness, originality, significance, and clarity, the relative weighting of these factors is not important.

**Key Questions For Authors:**

I don’t have any questions, as I found this paper to be very strong!

**Limitations:**

yes

**Strengths And Weaknesses:**

Strengths:
S1 (significance, originality): The paper’s motivation is an excellent one, building on both a longstanding debate in linguistics (about the autonomy of syntax) and connecting to recent advances in language model analysis. In building this motivation, the paper also does an excellent job of discussing relevant background literature.

S2 (soundness): The experiments are cleverly designed and show very clear and interesting results. It’s impressive how effectively these centroids can isolate syntax and semantics, and it’s impressive how much they can be disentangled.

S3 (soundness): I really appreciated the attention to detail in the experimental design, with many controls and follow-up analyses being run to ensure the validity of the design and the conclusions. E.g., right as I was starting to worry about potential syntactic ambiguity in the constructed sentences, the authors mentioned having anticipated that concern and having run an analysis to address it! Similarly, I also like the usage of ablating with a random sentence that is used in many of the experiments.

S4 (presentation): The paper is extremely clearly and carefully written.

S5 (significance): In addition to the results being interesting in their own right, it seems like this centroid-based approach could be useful for other researchers performing future research.

Weaknesses:
W1 (soundness): It’s possible that this experimental design somewhat biases the results toward syntax and semantics being disentangled. Specifically, the syntactic centroids are constructed from sets of sentences that are controlled to have very little in common in their meaning, while the semantic centroids are constructed from sets of sentences that are controlled to have very little in common in their form. Thus, the sentence selection might push toward disentanglement; it’s possible that other sentence templates could have pointed toward greater entanglement if, e.g., sentence structures were used that had stronger connections to meaning (I don’t have a clear proposal for what this would look like - but maybe using something from the construction grammar literature for cases where form and meaning are especially closely tied?) However, I see this more as a direction for future work than as a serious problem with the current work.

---

> ### Author Rebuttal · Authors · 2026-03-30
>
> We thank you for appreciating the interest of our research question and the robustness of our results.
>
> Concerning your suggestion to look at syntactic structures that have been shown to carry strong semantic information, that is indeed an interesting direction for future work. It is in line with what we are already finding (see discussion at the end of section 3.3). Namely, in very simplified terms, our results suggest that you can “remove” meaning and get an intact syntactic template, but if you remove the syntactic template, meaning is at least partially affected. In the current discussion, we link this to basic syntactic information, such as the relation of nouns with the main verb, being relevant to semantics, but it could also be due to construction-like semantic information being encoded in the syntactic template itself.

---

> > ### Author Rebuttal · Reviewer_9xMQ · 2026-04-03
> >
> > Thank you for the response! I have retained my high score.

---

> > > ### Author Response · Authors · 2026-04-06
> > >
> > > We thank the reviewer again for their positive evaluation of our work, its presentation and its impact.

---

### Official Review · Reviewer_GDZX · 2026-03-23

**Soundness:** 4
**Presentation:** 3
**Significance:** 4
**Originality:** 3
**Overall Recommendation:** 4
**Confidence:** 4

**Summary:**

This paper investigates how large language models (LLMs) encode syntactic and semantic information within their inner hidden layers. The authors extract "syntactic centroids" by averaging the representations of sentences that share the identical Part-of-Speech (POS) template but possess different meanings. Similarly, they extract "semantic centroids" by averaging representations across English paraphrases and translations into six distinct languages. By linearly subtracting these centroids from the sentence vectors, the authors demonstrate that syntax and semantics are, at least partially, linearly encoded. The study reveals a layer-wise processing pipeline where syntactic information remains highly salient across a wide range of layers, while semantic information peaks specifically in the central layers. Furthermore, the authors discover an asymmetric decoupling: removing semantic centroids does not significantly degrade syntactic similarity, but ablating syntactic centroids strongly negatively impacts semantic similarity.

**Compliance With Llm Reviewing Policy:**

Affirmed.

**Final Justification:**

> The authors engaged constructively during the rebuttal phase and provided interesting supplementary experiments (e.g., the early-training dynamics on Pythia-6.9b). I appreciate their transparency in acknowledging the limitations of their work.
>
> However, my core concerns regarding the methodological boundaries remain unresolved in the current manuscript. Specifically:
> 1. The operational definition of syntax relies purely on superficial POS sequences, ignoring the hierarchical parse trees and long-distance dependencies fundamental to structural syntax.
> 2. The empirical evaluation is heavily constrained to very short, simple sentences (6-10 words), leaving the robustness of the "asymmetric decoupling" claim untested on complex, multi-clause structures.
>
> Because the authors deferred these critical challenges to future work, I am retaining my score of 4 (Weak Accept). The paper presents a technically solid and thought-provoking preliminary investigation, but its overall impact and generalizability are currently limited by these experimental compromises. I recommend acceptance, but its priority should be calibrated against these limitations.

**Key Questions For Authors:**

1. Regarding the operational definition of syntax: POS tags capture superficial linear sequences but fail to capture hierarchical syntactic dependencies (e.g., nesting or long-distance agreement). How might your similarity and ablation results change if the "syntax twins" were matched based on exact syntactic parse trees rather than just POS sequences?
2. In Figure 5, the combined syntactic and semantic centroids explain only about 40% of the squared norm of the representations in the central layers. Do you have hypotheses regarding what specific types of information dominate the remaining 60% of this unexplained residual norm?
3. The dataset is explicitly constrained to relatively short sentences ranging from 6 to 10 words. Do you anticipate that the asymmetric decoupling of syntax and semantics will hold true for much longer sentences with highly complex, multi-clause structures?

**Limitations:**

yes

**Strengths And Weaknesses:**

Strengths:

1. The authors utilize a rank-based neighborhood similarity metric (Information Imbalance) rather than relying on linear metrics like CKA, which is well-justified for high-dimensional representation spaces.

2. The use of randomly permuted centroids as a control baseline effectively validates that the ablation specifically targets the intended linguistic features.
3. The empirical results are remarkably robust, having been replicated across three distinct model families and scales (DeepSeek-V3, Qwen2-7b, and Gemma3-12b).

Weaknesses:

However, a slight weakness in soundness is the shallow definition of syntax; equating syntax purely to linear POS templates ignores the hierarchical tree structures fundamental to true syntactic parsing.

---

> ### Author Rebuttal · Authors · 2026-03-30
>
> Thanks for acknowledging the soundness and robustness of our methodology and results, and for the constructive criticism.
>
> In the revision, we will explicitly clarify the points below.
>
> Q1:
>
> We chose POS sequences for practicality. In standard parsing formalisms, different POS sequences correspond to different syntactic structures, at least if labeled branches/dependency arcs are considered. Conversely, the same POS sequence can correspond to different syntactic structures. We manually checked that this was generally not the case with our structures (this is explicitly stated in footnote 1). We thus see our strategy as equivalent to one relying on a parser and grouping only sentences with the same syntactic structure. Comparing sentences with the same parse tree to sentences that share the same POS sequence but have different parses is an interesting experiment, but constructing the relevant data-set is challenging. First, it is likely that sequences with this property would often be ambiguous between the two possible parses, making it hard to group the data appropriately. Second, it is not clear that we could come up with  enough distinct POS sequences that have this property. We will mention this direction as future work.
>
> Q2:
>
> We agree that this is a very important point. As we will more clearly emphasize in the paper (this is currently briefly discussed at the end of Sec. 3.4 and in the limitations), we are not claiming that our syntactic and semantic centroids capture *all* of syntax and semantics, and we expect that part of the unexplained residual consists of syntactic/semantic dimensions we fail to capture. We conjecture moreover that the unexplained norm also includes lexical information about specific words (e.g., collocation information that can be useful to predict the next token, information about the orthographic make-up of the word, etc.), “meta-data” about, e.g., the language being used or style/register, as well as “algorithmic” information used by the model on the next layer (for example, to guide the attention mechanism, or induction heads). We are starting to think of ways to further partition the residual norms into these components, and we consider this a central direction for future research.
>
> Q3:
>
> As we acknowledged in our limitations, we constrained sentences to short lengths mainly due to LLM generation constraints, where even the state-of-the-art networks were showing many difficulties to generate high quality data following our prompts presented in Appendix A. Short sentences helped us to avoid a proliferation of possible parsing ambiguities, and to make sure all sentences and translations were carrying a sensible meaning that was easy to check. More fundamentally, to test our ideas about dissociating syntax and semantics inside vector representations, the natural point to start with is that of short, simple sentences. However, we fully agree with the referee, and we stated in the limitations, that future work should test the validity and or modifications of our results with longer sentences. We believe that the central challenge will be that of data generation. Given such a dataset, we conjecture that the asymmetric decoupling should hold true, given that one can still think of several long sentences sharing a fixed well-defined syntax structure while having completely unrelated meanings, while the syntax structure would be still fundamental to correctly compose each individual meaning. Nonetheless, it would be indeed very interesting to precisely quantify if, for example, the average token of long sentences is still able to retain syntactic information or if instead that information is washed away progressively with sentence length. We will add this discussion in the camera-ready version of our manuscript.
> To further investigate the asymmetric decoupling of syntax and semantics, we have now repeated our norm decomposition analysis on 5 Pythia-6.9b training checkpoints. Syntax appears to emerge first: very early in training, the syntax fraction of the squared norm averaged across layers has a peak, reaching 0.4 of the total norm. The corresponding semantics value is only 0.04. Later, the syntax fraction decreases, whereas the semantic component strongly increases, both converging to comparable values around 0.1. These preliminary results are aligned with our main findings, of syntax and semantics being encoded differentially, with asymmetric dependencies between them, and moreover suggest an intriguing asymmetry in learning. We will add them in an appendix.

---

> > ### Author Rebuttal · Reviewer_GDZX · 2026-04-05
> >
> > Thank you for the response! I have retained my score because most of my questions are listed as future works.

---

> > > ### Author Response · Authors · 2026-04-06
> > >
> > > We thank the referee for their constructive criticism. We believe the limitations present and explicitly acknowledged are not a weakness of our paper but genuine lines of interesting research to come, and we will make that clearer in the camera-ready version of our manuscript.

---

### Official Review · Reviewer_EQff · 2026-03-30

**Soundness:** 3
**Presentation:** 3
**Significance:** 2
**Originality:** 2
**Overall Recommendation:** 4
**Confidence:** 4

**Summary:**

This paper attempts to explore where and how syntactic and semantic information is stored in the layer representations of LLMs. They use a (motivate) methodology of subtracting the projection of the representation against a syntax or semantics vector, obtained by averaging multiple representations with the same syntactic structure or paraphrase semantics. That this works suggests that syntax and semantics are encoded (at least in part) by superposition (linear encoding). Syntax is more distributed, semantics is more in the middle layers; syntax can be separated from semantics but the reverse is harder.

**Compliance With Llm Reviewing Policy:**

Affirmed.

**Final Justification:**

The answers the authors give are reasonable and suggest good understanding. I still feel like my overall position (as in the initial review) is that this paper is only medium original and medium good not great. I will increase the rating to 4 but that seems about what it is worth to me.

**Key Questions For Authors:**

1. For averaged representations, is that of all tokens, or still only of the last N tokens (3 or 6)?
1. In Fig. 5, it seems somewhat worrying that under half of the vector content is either syntax or semantics in all cases. What does all the rest of the vector contain? Any leads or ideas?

**Limitations:**

Yes

**Strengths And Weaknesses:**

# Strengths

The methods used seem sound enough.

The paper is easy to read.

The motivations for subtracting the projection of the sentence vector along a calculated syntax or semantics vector seemed thought through and well motivated.

# Weaknesses

There is a certain arbitrariness in the methods used. Is concatenation of the last n tokens a valid representation. Averaging seems easier to motivate. Other work uses the last token only. Choosing a value of n > 1 seems harder to motivate.

I feel that the significance and originality of this paper is pretty limited. The results are roughly what you would expect and in line with what you would expect and results shown in previous work. For example, the fact that semantics is mainly in the middle layers is exactly the same result as in the Acevedo et al. (2025) paper that the authors cite.

---

> ### Author Rebuttal · Authors · 2026-03-31
>
> We thank the referee for acknowledging the soundness, the presentation and the motivations of our work. We will clarify in the revised version of our manuscript the key points discussed below in reply to the weaknesses and questions.
>
> Weakness 1, Representation arbitrariness:
>
> We believe that there is no consensus on what is the correct manner of representing a sentence, and different representations may be most adequate for different tasks, as our results seem to suggest.  Even if averaging may be a natural way to “summarize” a full representation of T tokens, it destroys positional information that may be relevant for some tasks. Moreover, for many other authors the last token is also a natural summary token in autoregressive models, and there is no fundamental criterion to choose one over the other.
> Concatenation of tokens was used here aiming to give importance to the positional information in the sentence, that is relevant for syntax similarity. We highlight that beyond Acevedo et al., Joshi et al. ([1] here) used the concatenation of the left and right boundaries of a token span, in the context of multi-token prediction.
> As it was also mentioned by other referees, we have added our results using the last token alone, which behaves similarly to the average and the concatenated representations, but tends to display a weaker signal.
>
> [1] Mandar Joshi, Danqi Chen, Yinhan Liu, Daniel S. Weld, Luke Zettlemoyer, and Omer Levy. SpanBERT:
> Improving pre-training by representing and predicting spans. Transactions of the Association for Computational Linguistics, 8:64–77, 2020. doi: 10.1162/tacl_a_00300. URL https://aclanthology.org/
> 2020.tacl-1.5/.
>
> Weakness 2, Significance and originality:
>
> As we stated in the introduction, the entanglement and or separability of syntax and semantics is an important and long-standing problem that doesn’t yet have a complete and definitive answer. We introduced a novel approach to study their encodings in LLMs with our syntax and semantic centroids, and we found an asymmetry in their coupling with the Information Imbalance, for which we gave an interpretation based on the independence of syntax. We are not aware of any literature that obtained an explicit differential representation for syntax and semantics information. This is what we do with our centroids. Acevedo et at. does not provide an explicit representation of the vector space encoding semantics, it only measures neighborhood alignment between translations. Moreover, it does not analyze syntax at all.
> Semantic information being expressed dominantly in central layers is not stated as a main novel finding anywhere in our work, and we indeed acknowledged that this was observed before.
>
> Question 1:
>
> To keep the comparison between representation choices fair, we indeed average the last N tokens (3 or 6). When it was possible, as for the II between paraphrases not using the semantic centroids, we did not observe significant differences in averaging 3 or 6 tokens.
>
> Question 2:
>
> We agree that this is a very important point. As we will more clearly emphasize in the paper (this is currently briefly discussed at the end of Sec. 3.4 and in the limitations), our syntactic and semantic centroids do not capture *all* of syntax and semantics, and we expect that part of the unexplained residual consists of syntactic/semantic dimensions we fail to capture. We conjecture moreover that the unexplained norm also includes lexical information about specific words (e.g., collocation information that can be useful to predict the next token, information about the orthographic make-up of the word, etc.), “meta-data” about, e.g., the language being used or style/register, as well as “algorithmic” information used by the model on the next layer (for example, to guide the attention mechanism, or induction heads). We are starting to think of ways to further partition the residual norms into these components, and we consider this a central direction for future research.
> We highlight however, that with only 2 vectors (S and T) out of the 7168 dimensions in DeepSeek-V3, explaining roughly 40 % of the norm of the activations is a non-trivial result on its own.

---

> > ### Author Rebuttal · Reviewer_EQff · 2026-04-07
> >
> > The answers the authors give are reasonable and suggest good understanding. I still feel like my overall position is that this paper is only medium original and medium good not great. I will increase the rating to 4 but that seems about what it is worth to me.

---

> > > ### Author Response · Authors · 2026-04-07
> > >
> > > We thank the referee for improving their score based on our answers. We are confident that our work is going to open several lines of research testing and extending the ideas that we presented, and that the intuitiveness of our protocol is going to facilitate its use by a wide audience.

---

### Decision · Program_Chairs · 2026-04-30

**Decision:**

Accept (regular)

**Comment:**

I want to assure the authors that I have carefully read the reviews, the rebuttals, and your confidential comments regarding reviewer engagement timing. These have all been fully factored into my final decision.

The authors present an elegant and computationally lightweight methodology—using "centroid subtraction"—to demonstrate that syntax and semantics are differentially encoded in LLMs. The core finding of an asymmetric decoupling (where ablating syntax degrades semantics, but not vice versa) is highly original and robustly validated across multiple modern models.

During the discussion, reviewers rightfully questioned the somewhat shallow operational definition of syntax (relying on POS sequences) and the evaluation's limitation to very short sentences. However, the authors engaged exceptionally well during the rebuttal. Crucially, they addressed these methodological doubts by providing compelling new experiments on Pythia-6.9b (showing syntactic representations emerge significantly earlier in pre-training) and proving pooling robustness with last-token experiments.

While extending this framework to complex, multi-clause structures remains a challenging direction for future work, the current paper is technically sound, non-redundant, and provides highly valuable insights into LLM interpretability. I recommend acceptance.